# DNA framework-engineered chimeras platform enables selectively targeted protein degradation

Li Zhou[1], Bin Yu[2], Mengqiu Gao[1], Rui Chen[2], Zhiyu Li[2,3], Yueqing Gu [1] ✉, Jinlei Bian [2,3] ✉ & Yi Ma [1] ✉

A challenge in developing proteolysis targeting chimeras (PROTACs) is the establishment of a universal platform applicable in multiple scenarios for precise degradation of proteins of interest (POIs). Inspired by the addressability, programmability, and rigidity of DNA frameworks, we develop covalent DNA framework-based PROTACs (DbTACs), which can be synthesized in high-throughput via facile bioorthogonal chemistry and self-assembly. DNA tetrahedra are employed as templates and the spatial position of each atom is defined. Thus, by precisely locating ligands of POI and E3 ligase on the templates, ligand spacings can be controllably manipulated from 8 Å to 57 Å. We show that DbTACs with the optimal linker length between ligands achieve higher degradation rates and enhanced binding affinity. Bispecific DbTACs (bis-DbTACs) with trivalent ligand assembly enable multi-target depletion while maintaining highly selective degradation of protein subtypes. When employing various types of warheads (small molecules, antibodies, and DNA motifs), DbTACs exhibit robust efficacy in degrading diverse targets, including protein kinases and transcription factors located in different cellular compartments. Overall, utilizing modular DNA frameworks to conjugate substrates offers a universal platform that not only provides insight into general degrader design principles but also presents a promising strategy for guiding drug discovery.

Protein degradation is an emerging strategy to treat diseases, especially proteolysis targeting chimeras (PROTACs)[1]. PROTACs are heterobifunctional molecules that comprise a ligand targeting the protein of interest (POI), an element that recruits E3 ligases, and a linker connecting the above two moieties[2–4]. Among them, the linker plays a critical role in bridging these two moieties. While many traditional linkers, including PEG[5], linear aliphatic chains[6,7], and more rigid (piperazine-type) linkers[5], have been studied, designing effective linkers remains challenging. Flexible linkers can cause configuration changes in PROTACs, leading to drug resistance[8]. Moreover,

structure–activity relationship studies (SARs) on linker lengths are primarily empirical and require time- and labor-intensive research[5]. In some instances, randomly selected linker lengths have been employed for PROTACs[6,9]. Therefore, identifying accurate and controllable protein degradation profiles necessitates critical consideration of desirable linker types, lengths, and even attachment points[10].

To date, improving the linker types has enabled the development of PROTACs from bivalent to trivalent, but the linker and its connection with the other three fractions pose significant synthetic burdens[11,12]. Moreover, even trivalent PROTACs have difficulties in

[1]Department of Biomedical Engineering, School of Engineering, China Pharmaceutical University, Nanjing 210009, China. [2]State Key Laboratory of Natural Medicines, China Pharmaceutical University, Nanjing 210009, China. [3]Jiangsu Key Laboratory of Drug Design and Optimization, Department of Medicinal Chemistry, China Pharmaceutical Universitys, Nanjing 210009, China. ✉e-mail: guengineering@cpu.edu.cn; bianjl@cpu.edu.cn; yima@cpu.edu.cn

efficiently conjugating multiple types of ligands, such as aptamers, antibodies, and peptides, or in simultaneously degrading different kinds of "undruggable" targets involving structural proteins, kinases, or transcription factors[13–16]. Thus, it is imperative to optimize degradation modes by introducing multiple ligands and targets simultaneously.

Recently, DNA has been engineered to form DNA frameworks[17], such as DNA tetrahedra[18], octahedra[19], and icosahedra[20], with well-controlled surface chemistry. The rigidity[21,22], addressability[23], and artificial programmability[24,25] of these DNA frameworks provide them with linker-like properties. The length of DNA frameworks can be precisely controlled with the spacing of two adjacent deoxynucleotides (~3.3 Å). Moreover, the caveolin-mediated endocytosis mechanism of DNA frameworks has been clarified[26], which is conducive to improving the poor cell entry efficiency of traditional PROTACs. Although bispecific aptamer chimeras[27], aptamer-PROTAC conjugates[28], RNA-PROTACs[29], and O'PROTAC[30] have been developed, the use of DNA frameworks as linkers has not been previously reported. We propose that the hybridization of DNA frameworks with PROTACs will provide multiple benefits: (1) High-throughput synthesis

of many variants will be possible through simple dynamic combinatorial chemistry and self-assembly, saving time and labor compared to the complicated synthesis of small-molecule PROTACs. (2) Precise positioning of ligands will be feasible due to the editability and ease of site-specific modification[31]. (3) The linker length can be easily controlled through DNA framework-engineered PROTACs. (4) A modular toolkit for rapidly creating highly selective and specific PROTACs will enable multi-target hydrolysis of different protein subtypes. (5) This approach will be ideally compatible with various ligands in the libraries, allowing for the degradation of different "undruggable" targets.

Inspired by the unique characteristics of DNA frameworks, we have developed an innovative strategy for the development of DNA framework-based PROTACs (DbTACs) by combining computational prediction, DNA self-assembly, and PROTACs technology (Fig. 1). To study linker length-activity relationships, we employ DbTACs formed with DNA tetrahedra, the ligands of cyclin-dependent kinase (CDK) family protein and cereblon (CRBN) E3 ligase as representative templates. Specifically, the position of the CDK9 ligand is fixed, while the CRBN ligand is shuttled on DNA tetrahedra to produce DbTACs with linker lengths ranging from 8 to 57 Å. DbTACs with different linker

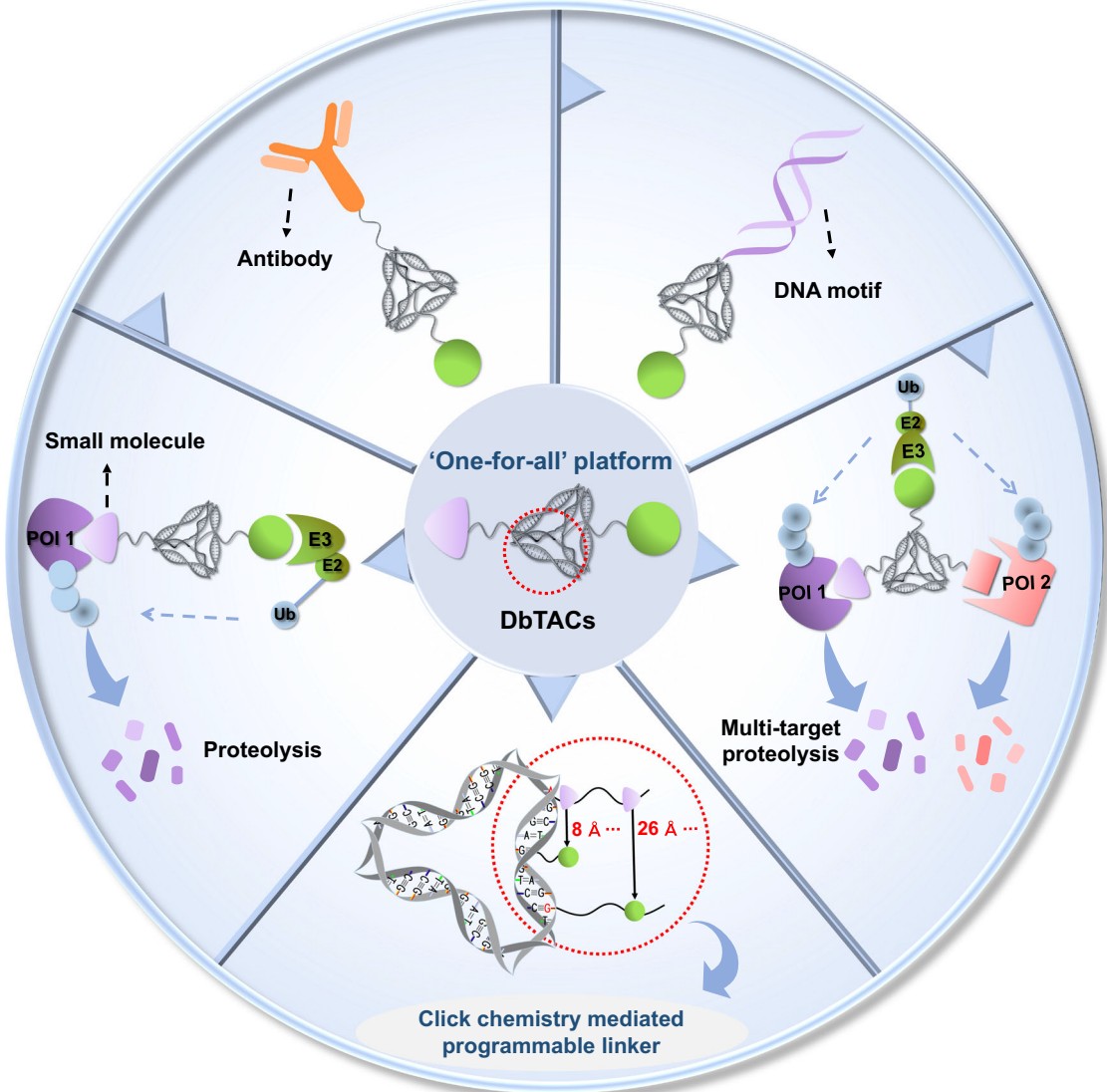

**Fig. 1 | Design of the DbTACs platform for selectively targeted protein degradation.** The schematic diagram of the DbTACs platform shows a click chemistry-mediated programmable linker. The universality of the platform is highlighted by the following features: (1) precise degradation of proteins of interest (POIs);

(2) simultaneous selective multi-target proteolysis; (3) compatibility with various warheads, such as small molecule, antibody, or DNA motif, for degrading different kinds of "undruggable" target proteins.

lengths are visualized, and an optimal DNA linker length of 26 Å is found to be the most effective in vitro. The difference in binding affinity explains the mechanism of linker length–activity relationships. Furthermore, we successfully demonstrate the feasibility of our idea by creating bispecific DbTACs (bis-DbTACs) sharing one CRBN ligand, which selectively degrades CDK6 (a protein involves in cell proliferation and differentiation) and CDK9 (a protein that participates in transcriptional regulation, DNA repair, and metabolism) in cancer cell lines. To expand the scope of our platform, antibody-, and DNA motif-based DbTACs are also developed that facilitate the degradation of CDK9 kinase and ETS-related gene (ERG) transcription factor. The information presented here has great potential to be applied to other DNA frameworks, known ligands, as well as targets in the libraries.

## Results

### DbTACs design, synthesis, and characterization

In this study, we developed a strategy for the design and construction of DNA framework-based PROTACs (DbTACs) by combining computational prediction and click chemistry-mediated programmable linker technology. We utilized the active ligands (L1 and L2) of CDK9 and CRBN that were previously investigated[32], which were subsequently modified with an azide group to serve as our models (Supplementary Fig. 1a). The distance between the two ligands was finely tuned by regulating the number of deoxynucleotides at a length-step of ~3.3 Å. DbTACs were designed and constructed with computer-aided technology to completely expose covalent ligands outside the DNA tetrahedra (Supplementary Fig. 1b). All DNA base sequences used for synthesizing DbTACs were presented in Supplementary Table 1. Herein, dibenzocyclooctyne (DBCO)-modified DNA strand S1 coupled with azide-modified L1 through click chemistry (Fig. 2a(i)), which was selected as a fixed design. Meanwhile, the azide-modified L2 sites on DBCO-modified S4 sequences were reasonably changed (Fig. 2a(iii)). After that, the above two DNA strands were self-assembled with DNA strands S2 and S3 to obtain various DbTACs with different linker lengths, including 8, 11, 16, 21, 26, and 57 Å (Fig. 2a(ii)). We specified a basic structural formula to define each compound using the term DbTACs-linker length, such as DbTACs-26 Å. The theoretically calculated linker lengths of DbTACs in the all-atom model match the prediction (Supplementary Fig. 1b). Successful DNA–ligand covalence was confirmed by molecular weight conjugation of mass spectrometry (Fig. 2b, c). The UV–visible spectra further showed characteristic absorption peaks of DNA, L1, and L2 (Supplementary Fig. 2a–g). According to Watson-Crick base pairing, four DNA sequences were then self-assembled into DbTACs by thermal annealing. The covalent attachment of ligands and step-by-step self-assembly of DbTACs was further confirmed via distinct bands with progressively delayed migration in agarose (Supplementary Fig. 3a) and polyacrylamide gel electrophoresis (PAGE) assays (Fig. 2d and Supplementary Fig. 3b). Moreover, the emission peak of representative DbTACs-26 Å overlapped L1 at 438 nm in fluorescence spectroscopy (Supplementary Fig. 2h). The green fluorescence of L2 was substantially quenched by DbTACs-26 Å (Supplementary Fig. 2i). The morphology of DbTACs-26 Å was clearly observed by atomic force microscopy (Fig. 2e). In conclusion, mounting evidence validates that DNA frameworks can be applied to bridge ligands. We have thus developed a strategy for simple and rapid preparing candidate DbTACs, which holds the potential for efficient protein degradation.

Considering the difficulty of visually characterizing site-specific ligands and linker lengths within DbTACs, single-strand DNA encoders containing polyadenine (polyA) of varying lengths (Supplementary Table 3) were employed to prepare overhangs at ligand-modified locations. These overhangs are capable of adsorbing Au nanoparticles (NPs) via electrostatic interaction[33–36]. Thus, Au NPs 5 nm (Supplementary Fig. 4a) and 10 nm (Supplementary Fig. 4b) were introduced to represent ligands L2 and L1, respectively. Furtherly, programmable

atom equivalents (Fig. 2f) were fabricated through self-assembly and easily observed by transmission electron microscopy (TEM). TEM images clearly revealed the distribution number and spacing of Au NPs in DbTACs equivalents (Fig. 2g). They were identical to the predesigned bivalent ligand modification and the angstrom-scaled linker length (average distance calculated to be 8 ± 2 Å (i), 11 ± 2 Å (ii), 16 ± 4 Å (iii), 21 ± 6 Å (iv), 26 ± 6 Å (v), and 57 ± 2 Å (vi)). Notably, these results were highly consistent with the linker lengths measured in all-atom models of DbTACs built using the aforementioned computer software. The precise control of linker lengths demonstrates the reliability and effectiveness of this method for ligand localization in DbTACs.

To ensure the viability of our in vitro evaluation, it was necessary to test the stability of DbTACs under physiological conditions. We found that intact DbTACs-26 Å remained visible in the PBS medium even after 24 h (Supplementary Fig. 5a, b). The half-life of DbTACs-26 Å in 10% FBS-contained medium was determined to be 29.2 h, and they remained stable for up to 6 h, suggesting an optimal incubation time for subsequent experiments (Supplementary Fig. 5c, d).

### DbTACs are highly potent CDK9 degraders

After establishing stable DbTACs, their degradation kinetics were investigated in live cells. We have reason to believe that water-soluble DbTACs can be efficiently internalized into cells through caveolin-mediated DNA tetrahedral endocytosis[26]. To investigate the fate of DbTACs after internalization, we conducted a subcellular localization assay using Cy3-labeled DbTACs-26 Å (DbTACs-26 Å-Cy3) in HepG2 cells. The cells were treated with DbTACs-26 Å-Cy3 for varying durations and subsequently stained with DAPI to track the intracellular localization of the compound. The results indicated that DbTACs-26 Å was effectively internalized by the cells and subsequently translocated to the nucleus (Supplementary Fig. 6). Therefore, we hypothesized that DbTACs with different linker lengths could be exploited to precisely control intracellular protein degradation. A human acute myeloid leukemia cancer cell line, MV4–11, was incubated with various DbTACs, and the degradation rates were analyzed using western blot (WB) (Fig. 3a). The results showed that DbTACs were excellent in inducing CDK9 degradation. Impressively, the degradation rate of DbTACs-26 Å at 200 nM achieved 75.2%, performing as well as the positive CDK9 targeting PROTAC B11[32]. In contrast, the degradation rates of DbTACs with linker lengths of 8, 11, 16, 21, and 57 Å were significantly lower than that of 26 Å, ranging from 18.2% to 65.2%. Importantly, a DNA tetrahedral control (tDNA) demonstrated that the DNA architecture alone was not responsible for the observed protein degradation. These trends were consistent with the degradation capacity of most degraders at 80 nM (Supplementary Fig. 7). Further immunofluorescence staining images (Fig. 3b) showed that DbTACs-26 Å induced weaker green fluorescence than the control and other tested DbTACs-linker lengths groups, indicating decreased expression of the CDK9 protein in this case. Therefore, the catalytic degradation activity of DbTACs could be programmed using DNA tetrahedral linkers. DbTACs sandwiched and consistently recruited the CRL4[CRBN] ubiquitin ligase complex to interact with CDK9 protein in a dose-dependent manner, with a $DC_{50}$ value of 95.15 nM (Fig. 3c). Representative DbTACs-26 Å downregulated CDK9 protein levels from 6 to 12 h (Fig. 3d). Although no significant protein degradation before 6 h, we cannot exclude the possibility that DbTACs-26 Å was undergoing cavitation-mediated endocytosis and ubiquitination[26]. The slight recovery after 24 h may result from counteracting factors such as protein resynthesis or the presence of feedback mechanisms[37].

To further investigate the efficacy of DbTACs in degrading CDK9 in live cells, we transfected human embryonic kidney cells (HEK293T) with plasmids encoding CDK9 fused with a localization signal (eGFP) and monitored protein abundance in real time. Successful introduction of exogenous CDK9 was confirmed by Supplementary Fig. 8a (i), and their gradual degradation was observed in HEK293T cells treated

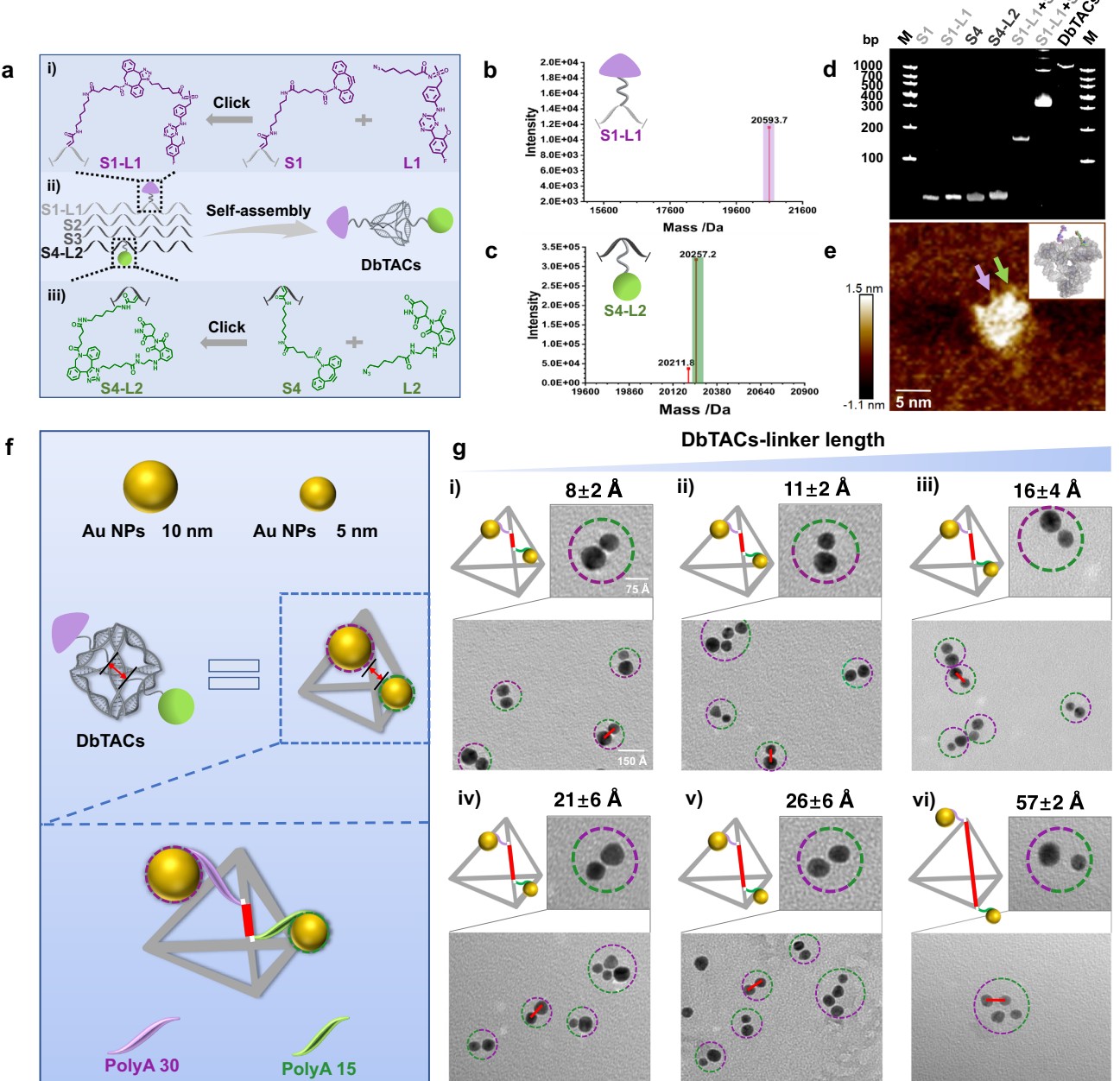

**Fig. 2 | Preparation and characterization of DbTACs. a** Chemical structures and design rationale of (i) S1-L1, (ii) DbTACs, and (iii) S4-L2 utilizing click chemistry and self-assembly. The mass spectra of **b** S1-L1 and **c** S4-L2. The theoretical molecular weight of S1-L1 (20,587.52 Da) = S1 chain (20,061.33 Da) + L1 (526.19 Da), and S4-L2 (20,250.38 Da) = S4 chain (19,795.19 Da) + L2 (455.19 Da). **d** PAGE analysis and **e** atomic force microscope image of representative DbTACs-26 Å. **f** Schematic illustration of the ligand covalent sites of programmable DbTACs equivalents to a DNA tetrahedral scaffold template with two polyA domains. The 5 nm and 10 nm Au NPs correspond to L2 and L1, respectively. **g** Cartoons and TEM images of prede-signed DbTACs-linker length equivalents obtained by adsorbing Au NPs to a DNA tetrahedral scaffold with two polyA domains. The scale bars, 75 Å and 150 Å, respectively.

with DbTACs for 6 h (Fig. 4a) and 12 h (Supplementary Fig. 8a (ii)) using inverted fluorescence microscopy. Flow cytometry analysis (Supplementary Fig. 8b) and WB analysis (Supplementary Fig. 8c) confirmed the depletion of CDK9-eGFP but not its control in HEK293T cells after 12 h. These results demonstrate the effectiveness of DbTACs in degrading CDK9 in live cells and suggest their potential utility in therapeutic applications.

**The enhanced binding affinity of the ternary complex**
We further explored the critical molecular mechanism underlying the ability of DbTACs to promote CDK9 degradation. Compared with free

DbTACs (7.158 min), size exclusion chromatography-high performance liquid chromatography (SEC-HPLC) results (Fig. 4b) revealed the shortest retention time (5.350 min) when DbTACs bound to both human recombinant CDK9 and CRBN proteins. This suggested that ternary complexes were efficiently formed. Molecular docking studies were then performed to predict the binding mode of DbTACs-26 Å with the protein binding sites (Fig. 4c). The modeling studies revealed that the ternary complex had a larger absolute docking score (−80.59) (Fig. 4c) compared to the binary complex (−54.54) (Supplementary Fig. 9). Furthermore, among investigated DbTACs with various linker lengths, DbTACs-26 Å displayed the most stable ternary conformation,

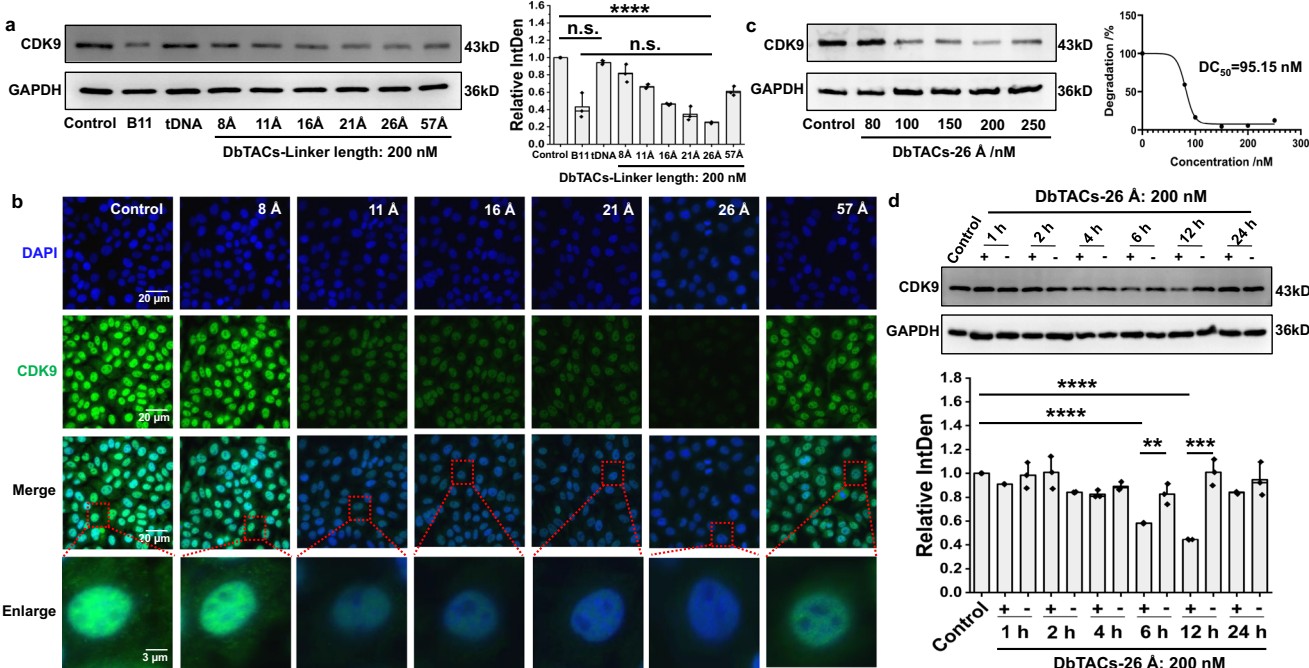

**Fig. 3 | Enhanced protein degradation mediated by DbTACs in cancer cells.**
**a** WB analysis and quantification of CDK9 protein levels in MV4–11 cells treated with 200 nM DbTACs with different linker lengths (DbTACs-8, −11, −16, −21, −26, and −57 Å) and a positive compound B11 for 6 h. GAPDH was used as a loading control. An unpaired two-tailed *t*-test was used to evaluate statistical significance. ****$P < 0.0001$ (Control vs. DbTACs-26 Å), n.s. represents no significance. The error bars indicate the mean ± SD values; $n = 3$. **b** Immunofluorescence staining images of human hepatoma cells (HepG2) treated with 200 nM DbTACs with different linker lengths or control for 6 h. The nuclei were stained with DAPI in blue, and CDK9 protein was stained in green. The red dotted square in the merged layer indicates an individual cell at a higher magnification. Scale bars, 20 and 3 μm, respectively. **c** Concentration-dependent degradation and **d** time degradation characteristics of CDK9 by representative DbTACs-26 Å analyzed by WB. GAPDH was used as a loading control. An unpaired two-tailed *t*-test was used to evaluate statistical significance. **$P = 0.0083$, ***$P = 0.0009$, ****$P < 0.0001$ (Control vs. 6 h+ and Control vs. 12h+). The error bars indicate the mean ± SD values; $n = 3$.

with a docking score of −80.59 (Supplementary Fig. 10). In contrast, the other linker lengths exhibited relatively lower docking scores: DbTACs-8 Å: −73.39, DbTACs-11 Å: −64.90, DbTACs-16 Å: −64.01, DbTACs-21 Å: −63.54, and DbTACs-57 Å: −75.63. This observation suggested that DbTACs-26 Å tended to form more stable ternary complexes. To further assess the binding kinetics of the ternary complex, surface plasmon resonance (SPR) assays were used. We immobilized CDK9 protein on a chip surface and measured the binding parameters of DbTACs alone (binary binding) or DbTACs preincubated with the CRBN protein partner (ternary binding). Multivalent effects may be responsible for the higher binding affinity of the CDK9-DbTACs-CRBN ternary complex ($K_D^{ternary} = 42.2$ nM) (Fig. 4d, Supplementary Fig. 11a) compared to the CDK9-DbTACs binary complex ($K_D^{binary} = 248.8$ nM) (Fig. 4e, Supplementary Fig. 11b) when excluding feeble interactions between CDK9 and CRBN (Supplementary Fig. 11c, d). The "positive cooperativity" ($\alpha = K_D^{ternary}/K_D^{binary}$, $\alpha > 1$) thus further stabilizes the ternary complexes, allowing for the desired biological effects. Importantly, DbTACs-26 Å ($K_D^{ternary} = 42.2$ nM) exhibited a robust binding affinity with CDK9 and CRBN proteins, in contrast to DbTACs-8 Å ($K_D^{ternary} = 185.4$ nM) (Fig. 4f, Supplementary Fig. 11e) and DbTACs-57 Å ($K_D^{ternary} = 120.7$ nM) (Fig. 4g, Supplementary Fig. 11f). This indicates that the enhanced binding affinity of DbTACs-26 Å is due to the appropriate spacing provided by the 26 Å linker length, allowing two ligands to match their corresponding sites and forming a stable ternary complex that facilitate the degradation of CDK9.

### The selectivity and degradation mechanism of DbTACs
To better understand the mechanism behind the selective degradation induced by topologically engineered DbTACs, we investigated their activity towards specific proteins. DbTACs-26 Å demonstrated selective degradation activity toward CDK9 while exhibiting no potency

towards CDK1/2 and CDK6 (Fig. 4h). Conversely, the positive control drug B11 exhibited non-selective specificity towards CDK6 and CDK9. Thus, like a "dumbbell", DbTACs retained high selectivity in inducing specific protein–ligand and protein–protein contact. This could be attributed to the high stability and rigidity of DNA tetrahedra, which differs from traditional linear linkers. To further scrutinize the pathway of CDK9 degradation induced by the selective modulator, several rescue experiments were carried out. As expected, the excessive monovalent ligands BAY-1143572 (BAY.) or pomalidomide (P.M.) effectively blocked the active regions of the kinase and ligase, thereby impeding the function of DbTACs-26 Å (Fig. 4i, Supplementary Fig. 8d). In this case, the degradation of CDK9 was also hindered in cells pre-treated with MG132, a proteasome inhibitor. The results confirmed that the mechanism of DbTACs involves effectively binding to CDK9 and Cullin RING ligase, as well as proteasome-mediated degradation. To investigate the mRNA levels of *CDK9* and downstream *MCL-1* (an antiapoptotic gene), RT-qPCR assays (Supplementary Fig. 12) were further performed. Importantly, there were no significant differences in mRNA levels between treated groups and control groups, reinforcing the mechanisms verified above. Having proven reductions in disease-associated CDK9 protein via the proteasomal pathway, DbTACs-26 Å led to substantially more potent cell apoptosis (Supplementary Fig. 13) and cytotoxicity (Supplementary Fig. 14) than the control group. To evaluate the level of autophagy, we employed a fluorescence-based assay using an Autophagy Staining Assay Kit with MDC (Supplementary Fig. 15). After MV4–11 cells were treated with DbTACs-26 Å at different concentrations, the mean fluorescence intensity of the MDC probe was significantly increased compared with the control group, indicating the accumulation of acid compartments. In particular, the fluorescence intensity of the 200 nM DbTACs-26 Å was similar to that of the autophagy-induced positive group, indicating

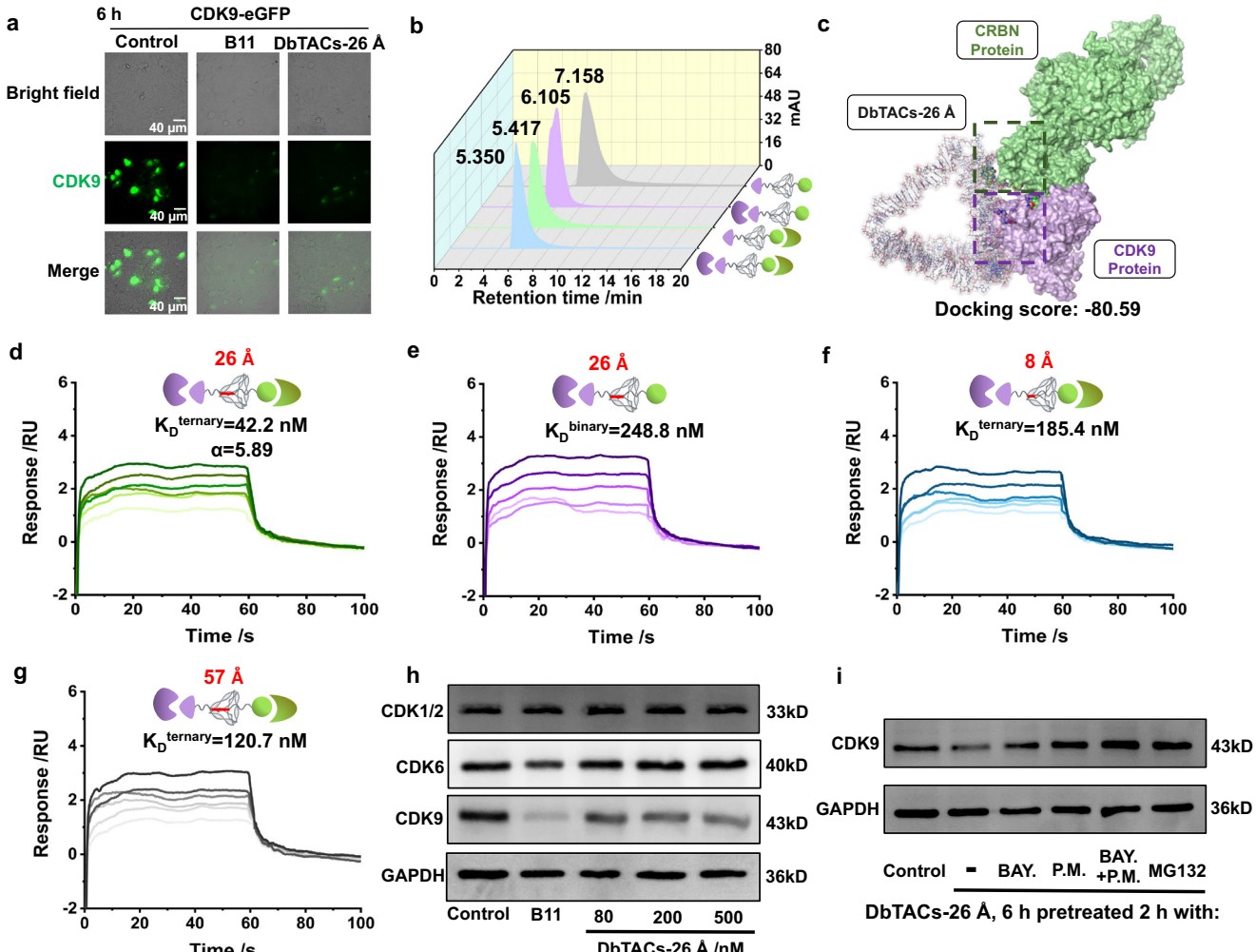

**Fig. 4 | Real-time visualization of protein degradation and mechanism of DbTACs. a** Live-cell imaging was performed to visualize the real-time localization of CDK9 in HEK293T cells and to track the decrease in CDK9 after treatment with DbTACs-26 Å for 6 h. The scale bars, 40 μm. **b** SEC-HPLC analysis of retention time of DbTACs-26 Å after incubation with human recombinant CDK9 or CRBN protein or both. **c** Molecular docking sites of the ternary complex in an all-atom model. SPR sensorgrams were employed to monitor the interaction between **e** DbTACs-26 Å (binary complexes) or **d** DbTACs-26 Å, **f** DbTACs-8 Å, and **g** DbTACs-57 Å preincubated with human recombinant CRBN protein (ternary complexes) and immobilized CDK9 protein. **h** WB analysis of the selective degradation potency of DbTACs-26 Å in MV4–11 cells. The cells were co-incubated with different treatments and collected after 6 h. **i** A ligand competition test for the degradation of the target protein by CDK9 degrader DbTACs-26 Å. Inhibitors of CDK9, CRBN, and proteasome (BAY., P.M., and MG132, respectively) were used, and all signals of each band were normalized successively to GAPDH.

that autophagosomes were formed during autophagy. Together, these findings demonstrate that DbTACs promote autophagy in our cellular model. Collectively, the biological data support DbTACs for valid protein degradation and downstream intervention.

To further investigate the selective degradation mechanism of DbTACs, 4D-FastDIA quantitative proteomic analysis was performed on the MV4–11 cells from different treatment groups. Based on these qualified data (Supplementary Fig. 16), a total of 43,504 peptides, 6294 proteins, and 6275 quantifiable proteins were identified in MV4–11 cells. Principal component analysis (PCA) revealed distinct protein expression patterns between the control group and the DbTACs-26 Å group, and they were relatively separated from each other (Supplementary Fig. 17). In the proteins exhibiting significant changes, the DbTACs-26 Å group displayed 11 upregulated proteins and 131 downregulated proteins (Fig. 5a). Notably, among the down-regulated proteins, CDK9 protein exhibited the most significant downregulation in abundance, while CDK1/2 and CDK6 proteins were not downregulated. These results are consistent with the aforementioned cellular experiments. To investigate the potential functional enrichment of the differentially expressed proteins, subcellular

distribution analysis (Fig. 5b) and cellular component analysis (Fig. 5c) were performed on these proteins. The results indicated prominent localization of these differentially expressed proteins in the nucleus, supporting the targeting of nuclear proteins by DbTACs-26 Å. Further functional analysis, including gene ontology (GO) enrichment analysis, highlighted the involvement of differentially expressed proteins in critical biological processes related to DNA damage and cell cycle processes (Fig. 5d). Importantly, the molecular function analysis emphasized the impact on cyclin-dependent protein kinase activity, aligning with targeted degradation of CDK9 by DbTACs-26 Å (Fig. 5e). Furthermore, the KEGG pathway analysis revealed the p53 signaling pathway as a key mechanism associated with the significantly changed proteins in the DbTACs-26 Å group (Fig. 5f). Protein–protein interaction analysis further demonstrated interactions among different proteins, including CDK9, and their association with TP53, supporting the selective degradation mechanism of DbTACs-26 Å (Fig. 5g).

## Multitargeted proteolysis of DbTACs
Informed by the preparation method of DbTACs, a modular trifunctional agent, named bispecific DbTACs (bis-DbTACs), was generated to

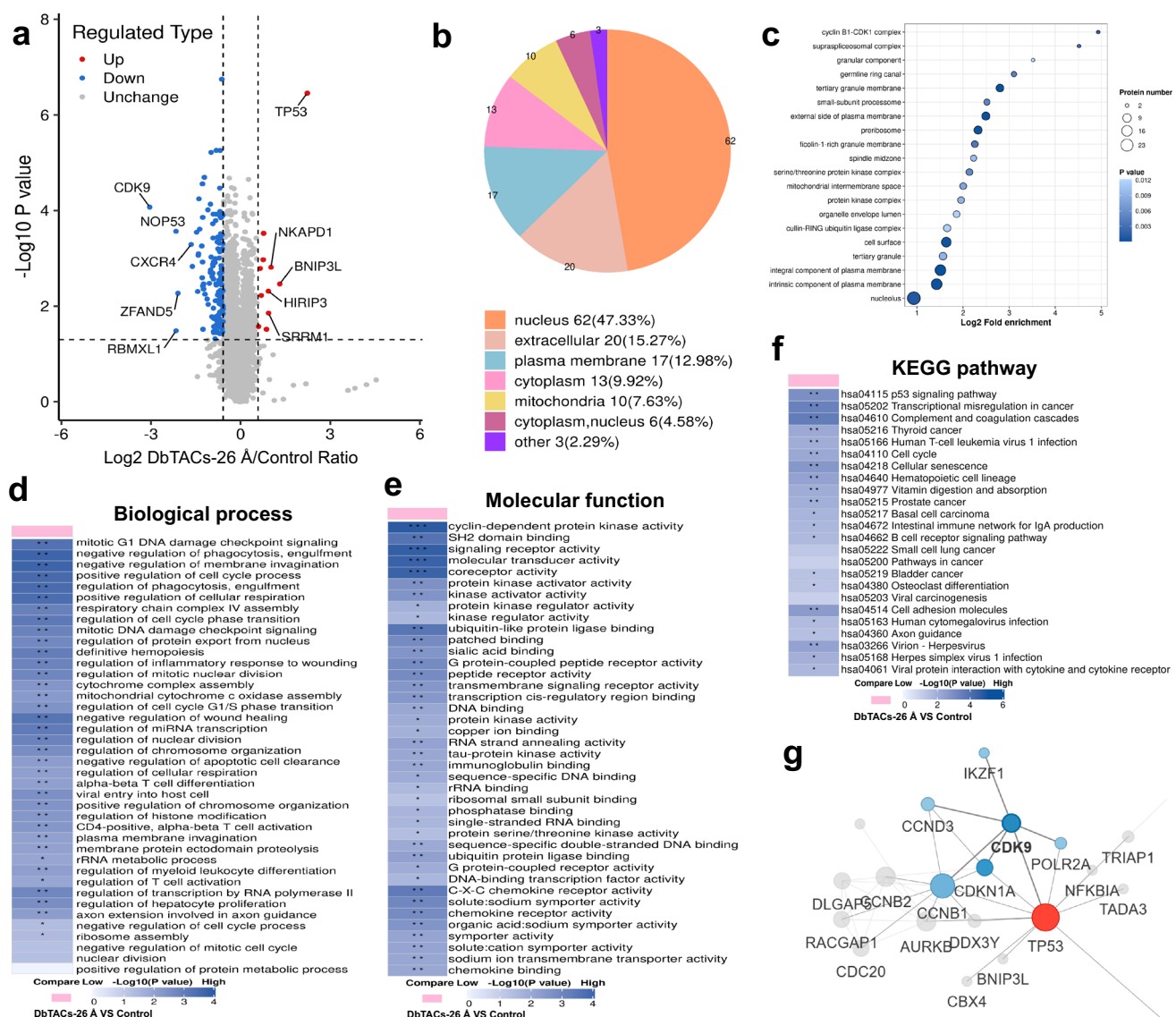

**Fig. 5 | Proteomics analysis of differential protein in MV4–11 cells treated by DbTACs-26 Å. a** Volcano plot showing fold changes of protein abundance from global proteomics analysis of MV4–11 cells treated with DbTACs-26 Å for 6 h at 200 nM. Statistical test (*t*-test analysis). **b** Subcellular localization prediction of identified proteins using WoLFPSORT. The subcellular localization of identified proteins was predicated using the WoLFPSORT database with amino acid sequences of identified proteins. **c** Gene ontology (GO) analysis of a cellular component of samples between DbTACs-26 Å-treated group and control group. Statistical test (Fisher's exact test). Molecular function analysis of compound DbTACs-26 Å-treated group and control group. Cluster analysis of **d** biological process, **e** molecular function, and **f** KEGG pathway of samples between DbTACs-26 Å-treated group and control group. *n* = 3. Statistical test (Fisher's exact test). **g** Protein–protein interaction network analysis of CDK9 with other proteins.

extend this platform. Herein, a covalent ligand 3 (L3) was designed to target CDK6 protein and shared one E3 ligase ligand (L2) with CDK9 ligand (L1) on the DNA tetrahedral template (Fig. 6a). Similarly, an all-atom model of bis-DbTACs was initially constructed to observe its configuration (Supplementary Fig. 18). The ligands loading and the establishment of bis-DbTACs were confirmed by the presence of a single band and slow migration in gel electrophoresis (Supplementary Fig. 19a, b) and the characteristic peaks in UV–visible spectra (Supplementary Fig. 19c). We also patterned 5, 10, and 15 nm Au NPs (Supplementary Fig. 4c) with valence bond analogs using multiple sticky polyA domains, to generate tandem structures of three Au NPs with different diameters, corresponding to L2, L1, and L3 based on molecular sizes (Fig. 6b). TEM images showed the successful generation of the desired structures (Fig. 6c). Surprisingly, bis-DbTACs exhibited concentration-dependent consumption in CDK6 levels, while retaining a 55% degradation rate of CDK9 (Fig. 6d). Notably, the

individual tDNA component was not responsible for the degradation of CDK6 or CDK9. Furthermore, neither of bis-DbTACs nor tDNA induced degradation of CDK1/2. This phenomenon is primarily due to target selectivity and differences in ligand–protein affinities. Finally, double-immunofluorescence studies of HepG2 cells showed that bis-DbTACs markedly attenuated CDK6 (red) and CDK9 (green) fluorescence compared to the control group (Fig. 6e, Supplementary Fig. 19d). To assess the impact of bis-DbTACs on the cell cycle, we performed flow cytometry analysis employing propidium iodide (PI) staining (Supplementary Fig. 20). After treating MOLM13 cells with bis-DbTACs, a significant reduction in the proportion of cells in the S phase, which is critical for DNA replication, was observed when compared to the control group. This finding implied that bis-DbTACs possess the ability to impede DNA synthesis, thus modulating the cell cycle progression. In summary, these data suggest that bis-DbTACs are effective degraders of CDK6 and CDK9, providing a basis for their

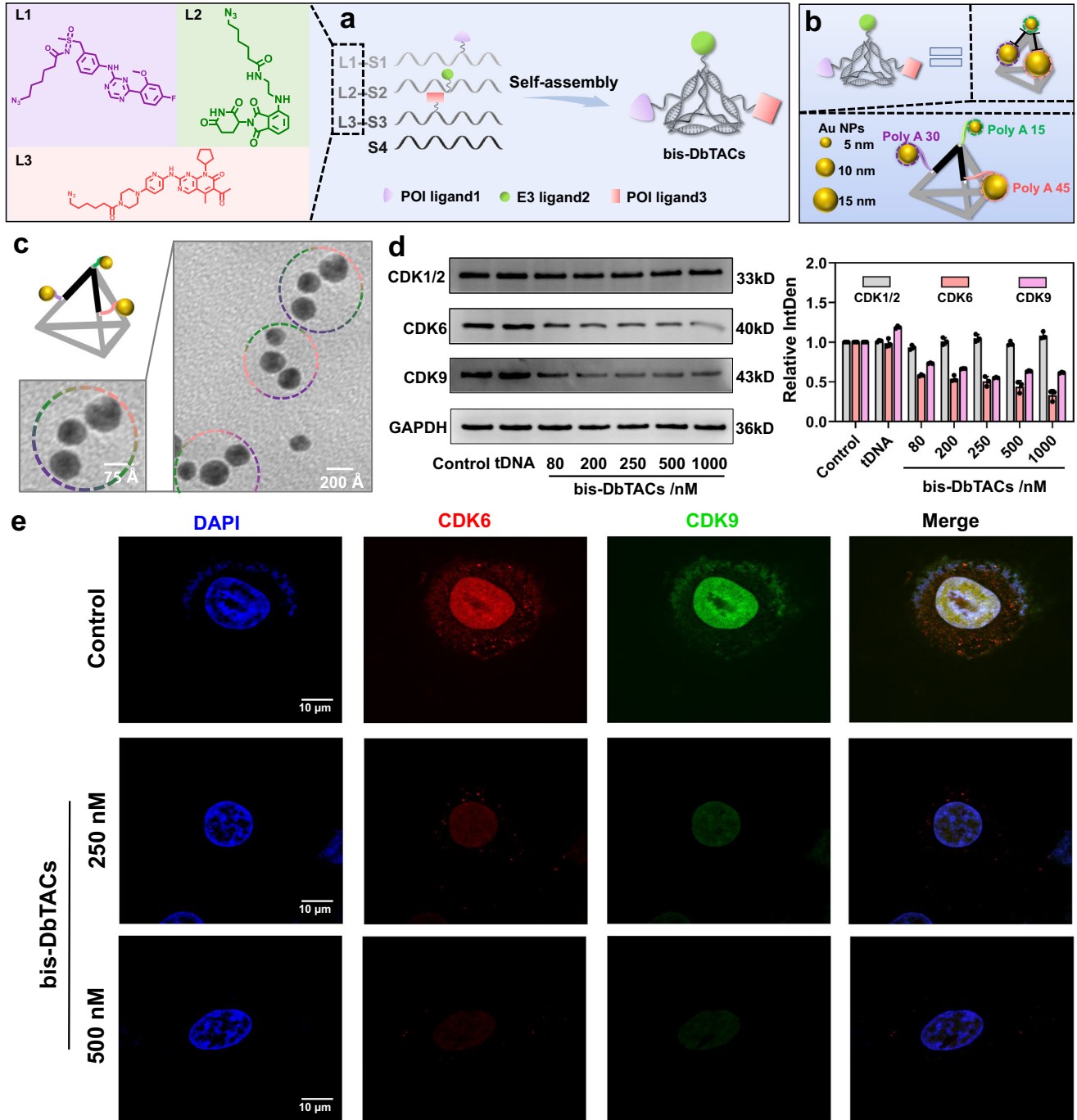

**Fig. 6 | Design, preparation, characterization, and efficacy of bis-DbTACs.**
**a** Schematic illustration of bis-DbTACs design, which is based on DbTACs. **b** Schematic illustration of three ligand covalent sites of bis-DbTACs equivalent to a DNA tetrahedral scaffold with three polyA domains. Au NPs (5, 10, and 15 nm) correspond to CRBN, CDK9, and CDK6 ligands, respectively. **c** Cartoon and representative TEM images of bis-DbTACs equivalents. Scale bars are 75 Å and 200 Å, respectively. **d** WB analysis of the selectively targeted degradation ability of

bis-DbTACs at different concentrations and semiquantitative analysis of the grayscale. The error bars indicate the mean ± SD values; *n* = 3.
**e** Immunofluorescence double-staining images of HepG2 cells treated with/without bis-DbTACs were recorded by confocal laser scanning microscopy. The cell nucleus was stained with DAPI. CDK6 and CDK9 proteins were labeled with anti-CDK6 and anti-CDK9 antibodies, respectively. Scale bars, 10 μm.

potential applications in cancer treatment. Moreover, this strategy holds promise for the development of degraders targeting dual or multiple targets.

### Generalizability of DbTACs
Many proteins lack known small-molecule ligands, so we focused on whether antibodies (Abs) could be used as target ligands since they are more broadly available for "undruggable" proteins. Firstly, CDK9 Abs

and L2 were selected and conjugated onto DNA tetrahedra to prepare Abs-based DbTACs (Abs-DbTACs), further providing the universality of the platform. We coupled commercially available Abs that recognize CDK9 full length (1–372 aa) to 5′-thiol-modified DNA tetrahedra through the heterobifunctional crosslinker N-succinimidyl 3-maleimidoproppionate (BMPS) (Fig. 7a). After purifying Abs-crosslinker covalent via NAP-5 desalting column, the absorbance of the product was redshifted from 465 nm to 595 nm by the Bradford assay

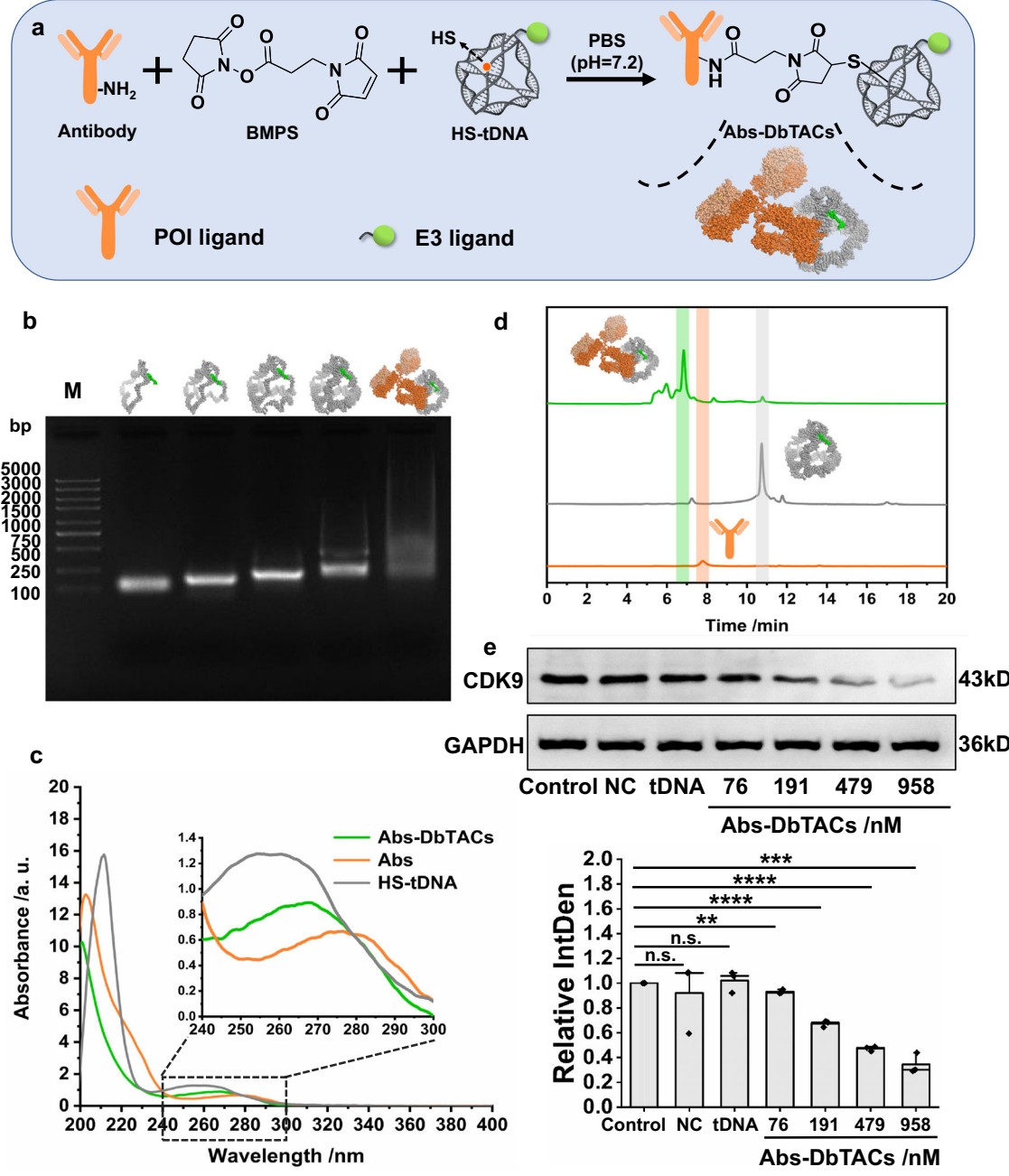

**Fig. 7 | Design, preparation, characterization, and efficacy of Abs-DbTACs formed using antibody as POI ligand. a** Strategy for designing Abs-DbTACs using CDK9 antibody as the POI ligand. **b** Self-assembly process of Abs-DbTACs was analyzed by agarose gel electrophoresis. The preparation of Abs-DbTACs was verified by **c** UV–visible spectra and **d** SEC-HPLC. **e** WB analysis of the targeted CDK9 degradation ability of Abs-DbTACs at different concentrations in MOLM13 cells. A free CDK9 antibody was chosen as the negative control (NC). A semi-quantitative analysis of their grayscale values was performed. The error bars indicate the mean ± SD values; $n = 3$. An unpaired two-tailed $t$-test was used to evaluate statistical significance. **$P = 0.0016$, ***$P = 0.0002$, ****$P < 0.0001$ (Control vs. 191 nM Abs-DbTACs and Control vs. 497 nM Abs-DbTACs), n.s. represents no significance.

(Supplementary Fig. 21), indicating the potential for subsequent coupling with sulfhydrylated DNA tetrahedra. Agarose gel results further confirmed that DNA tetrahedra were conjugated to the Abs (Fig. 7b). The characteristic UV–visible absorption peak of Abs-DbTACs was 267 nm, located between HS-tDNA (260 nm) and Abs (280 nm) (Fig. 7c). Abs-DbTACs exhibited remarkable stability in PBS for up to 12 h (supplementary Fig. 22a) and in cell medium for up to 6 h (supplementary Fig. 22b), and showed resistance to protease (supplementary Fig. 22c) and DNase I (Supplementary Fig. 22d) degradation within 12 h. Further, SEC-HPLC (Fig. 7d) analyzed that the molecular weight largest Abs-DbTACs exhibited the shortest retention time

(6.490 min). The Abs (7.790 min) and DNA tetrahedra (10.734 min) used alone, however, were easily intercepted by the gel chromatographic column due to their relatively small molecular weights. Additionally, the concentration of Abs-DbTACs in the mixture was quantified to 9.85 μM according to the peak area ratio (23: 1) of Abs-DbTACs to HS-tDNA. It is worth noting that Abs-DbTACs were prepared without site specificity, and different lysine residues may be modified in this study. Therefore, Abs-DbTACs herein were not a single peak. We hypothesize that this may be beneficial for recognition, as the mixture of binders can be delivered in different directions to recruit more POIs and E3 ligase. To further study antigen-antibody binding

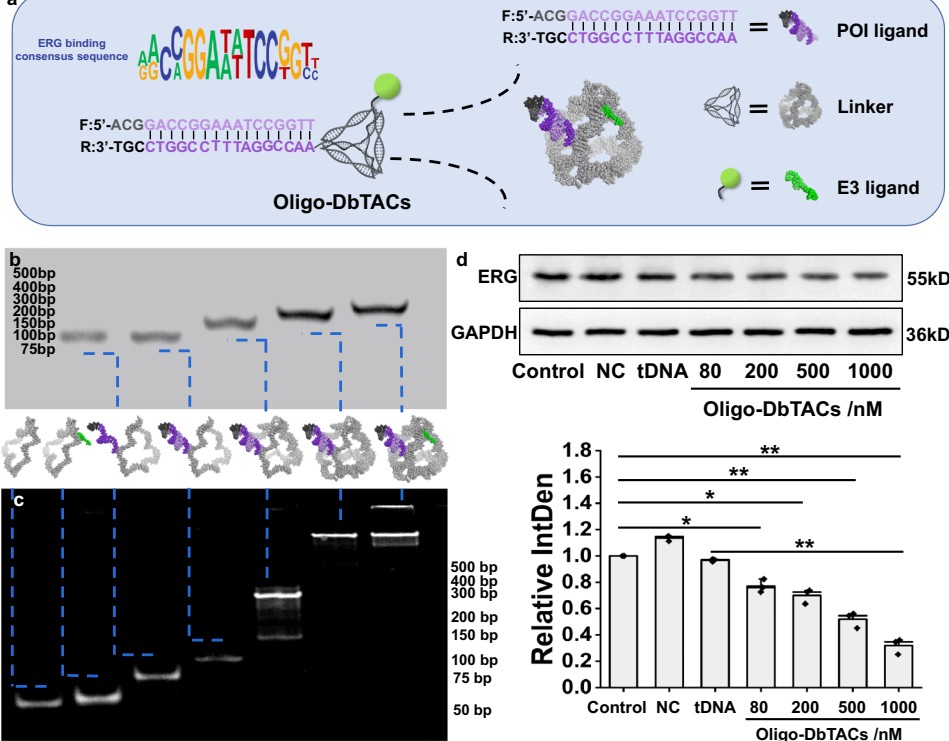

**Fig. 8 | Design, preparation, characterization, and efficacy of Oligo-DbTACs formed by DNA motif as POI ligand. a** Schematic illustration of the design strategy for Oligo-DbTACs formed using DNA tetrahedra as a linker conjugating L2 and ERG ligand, with binding moiety (purple) and three extra nucleotides (black) for protection of oligo degradation. **b** Agarose gel electrophoresis and **c** PAGE images of the self-assembly process of Oligo-DbTACs formed using DNA motif as POI ligand. **d** WB analysis of the targeted degradation of ERG protein by Oligo-DbTACs with different concentrations in PC3 cells. A non-specific sequence was chosen as the negative control (NC) target ligand. And the semiquantitative analysis of their grayscales. The error bars indicate the mean ± SD values; $n = 3$. A paired two-tailed t-test was used to evaluate statistical significance. *$P = 0.0153$ (Control vs. 80 nM), $P = 0.0111$ (Control vs. 200 nM), **$P = 0.0051$ (Control vs. 500 nM), $P = 0.0025$ (Control vs. 1000 nM), and $P = 0.0030$ (tDNA vs. 1000 nM).

properties, WB data (Fig. 7e) confirmed a significant 65.5% decrease of CDK9 expression in MOLM13 leukemia cells following a 12 h treatment with Abs-DbTACs. Similarly, the individual tDNA exhibited negligible influence on the degradation process. Overall, these results demonstrate that antibodies are preferable for DbTACs to degrade proteins without known small-molecule ligands.

To further illustrate the "one-for-all" properties of DbTACs platform, different types of ligands and targets were investigated simultaneously. Specifically, a DNA motif that specifically recognizes the ETS-related gene (ERG), a highly cancer-related transcription factor, was used as a ligand. We designed and synthesized oligonucleotide-based DbTACs (Oligo-DbTACs) by introducing DNA tetrahedra as a linker conjugating L2 and ERG ligand, with binding moiety (purple) and three extra nucleotides (black) for protection of oligo degradation, according to literature modification[30] (Fig. 8a). A non-specific sequence (ACGCGATCGAGATGTACTT) was chosen as a negative control (NC) based on in silico prediction by the PROMO software, ensuring that no known proteins would bind to this sequence (Supplementary Fig. 23). Mass spectrometry showed that the DNA strand successfully loaded the reverse ERG binding consensus sequence (Supplementary Fig. 24). The successful construction of Oligo-DbTACs was confirmed through the observation of significant delays in electrophoretic mobility shift assay (Fig. 8b, c). These Oligo-DbTACs were delivered into PC-3 prostate cancer cells, which overexpressed ERG. The kinetics experiment confirmed that Oligo-DbTACs, but not the NC or tDNA, effectively degraded ERG protein in a dose-dependent manner (Fig. 8d). In summary, these results suggest that DbTACs derived from oligonucleotide ligands have the potential to effectively degrade various "undruggable" targets, such as transcription factors, DNA and RNA binding proteins.

In addition, we explored the degradation of proteins located in different cellular compartments, including cytoplasmic hematopoietic progenitor kinase 1 (HPK1), which holds great promise for cancer immunotherapy. Thus, we developed HPK1-DbTACs that utilized a DNA tetrahedral scaffold co-conjugated with the azide-modified HPK1 ligand (L4) and CRBN ligand (L2). Through WB analysis (Supplementary Fig. 25a, b), a concentration-dependent degradation of HPK1 by these HPK1-DbTACs was observed, in contrast to the HPK1 kinase inhibitor sunitinib, which showed no degradation. Importantly, negative controls (Control, tDNA, tDNA-L2, and tDNA-L4) did not induce degradation either. Furthermore, the $DC_{50}$ fit of HPK1 degradation was determined to be 262.0 nM (Supplementary Fig. 25c). These findings demonstrated the potential of DbTACs to target and degrade cytoplasmic proteins, expanding the application of the DbTACs platform.

## Discussion

The research presented here takes inspiration from a commonly used Chinese proverb, "all for one, and one for all", attempting to develop a modular platform for PROTACs. For this purpose, we outline a highly modular linker design strategy that utilizes framework nucleic acid as scaffolds, to which both target ligands and E3 binders can be conjugated. As an advantageous platform, DbTACs afforded resolution at the angstrom level for interrogating spatial distance-activity relationships. Indeed, further mechanistic experiments showed that representative DbTACs-26 Å exhibit high binding affinity, with $K_D$ values that are 2.86- to 4.39-fold lower than those of DbTACs-8 Å and DbTACs-57 Å. Given this advantage, we are capable of precisely designing DNA

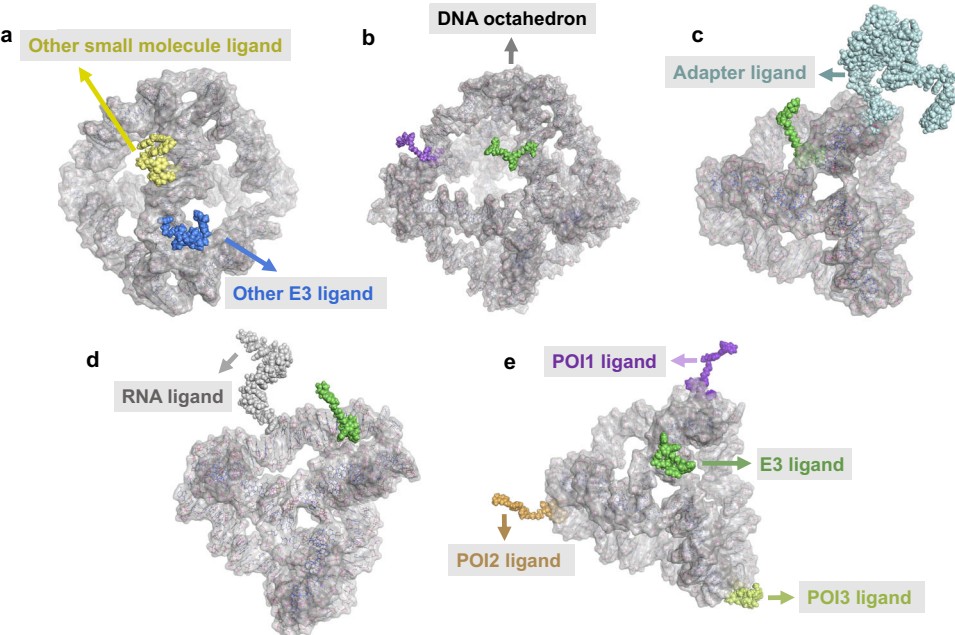

**Fig. 9 | The potential of the DbTACs platform based on computational tools prediction.** All-atom models of DbTACs for coupling with **a** other small molecule ligands and E3 ligands created using PolygenDNA and MOE software. **b** A DNA octahedron-based DbTACs model for exploring the diversity of DNA frameworks. All-atom models of DbTACs for coupling with **c** aptamers and **d** RNA chains as pharmacophores built using PolygenDNA and MOE software. **e** All-atom model of DbTACs for simultaneous selective degradation of multiple targets.

linker lengths and manipulating degradation. While DbTACs carry the risk of off-target effects, their modularity and potential for selective protein targeting make them a promising strategy for protein degradation. Continued refinement and optimization of DbTAC design principles will enable researchers to harness their full potential, advancing research of DNA-binding proteins and facilitating the development of targeted therapies.

Additionally, the transition from conventional bivalent to trivalent PROTACs[12,38] may not be a beneficial approach to immediately improve multi-target degradation, given poor water solubility and complex chemical synthesis. On the one hand, the DbTACs platform addresses poor water solubility. They were primarily based on DNA sequences, which are water-soluble molecules. They can be easily dissolved in aqueous solutions or PBS without the need for an organic solvent. As a result, DbTACs do not require solvent replacement during the experimental process. This feature simplifies the experimental procedures and reduces potential side effects associated with residual organic solvents. On the other hand, water-soluble DbTACs can be easily synthesized via high-throughput click chemistry, as well as other methods such as Staudinger ligation chemistry[39] and activated esters[40]. Moreover, the selectivity of DbTACs enables precise degradation of multiple targets. We have demonstrated herein that DNA tetrahedral framework-engineering chimeras can be used to deal with proteins and initiate numerous future applications. For example, DbTACs formed with antibodies (Fig. 7) or DNA motifs (Fig. 8), or other types of elements as ligands have shown a powerful role in protein degradation. There is scope for prospective variations in DbTACs technology, exemplified by using different E3 ligase (CRBN, VHL, MDM2, etc.) ligands (Fig. 9a) and ideal DNA framework (DNA octahedron, icosahedron, etc.) (Fig. 9b). If various types of target ligands are known (small molecules, aptamers, RNA, polypeptides, nanoantibodies, etc.) (Fig. 9a, c, d), this would allow all warheads to engage under structural constraints. However, to fully understand the potential, versatility, and generalizability of DbTACs technology, a more diverse range of target proteins and animal studies should be investigated in the future.

In summary, we have proposed DNA framework-engineered DbTACs with programmable linkers as an approach for the development of empirical nature in the linker domain. Compared to conventional PROTACs, DbTACs present a modular approach to generating degraders. This approach effectively reduces the number of steps involved in preparing the linker itself, as well as the subsequent coupling of the linker with the ligands. This represents a significant step forward in the development of more efficient drug production methods. We anticipate that DbTACs will serve as a promising platform for researchers to screen ligand candidates in the future. Furthermore, we expect the usage of our platform as a chemical biology tool to continue interdisciplinary collaboration and the innovation of other bi-, tri-, and even multi-specific drug models (Fig. 9e). We look forward to further validation of this exciting pharmacological modality.

## Methods
### Chemical synthesis
Details of the synthesis of the compounds and their intermediates are provided in the Supplementary Note. MestReNova (v12.0.3) was used for all chemical NMR analyses.

### Equipment
Mass spectra of the samples were acquired using an LTQ XL liquid mass ion trap mass spectrometer (Thermo, USA). The atomic force microscope image was taken by Burker AXS Dimension Icon. The particle sizes of the Au NPs were determined using a Litesizer 500 Particle Size Analyzer (Anton Paar, Austria). The UV–visible spectra were recorded with a NANODROP ONE (Thermo Fisher Scientific, USA). Fluorescence spectra were obtained on a fluorescence spectrometer (Sartorius, Germany).

### Fabrication of DbTACs and HPK1-DbTACs
The DNA strands were purchased from Sangon Biotech Co., Ltd. (Shanghai, China). The DNA sequences used are shown in Supplementary Table 1. To form DbTACs and HPK1-DbTACs, the single-stranded DNA S1 (50 μM) was mixed with CDK9 ligand (500 μM) or

HPK1 ligand (500 μM) at a 1:1 molar ratio. Additionally, DNA S4 (50 μM) was mixed with CRBN ligand (500 μM) at a 1:1 molar ratio. The two solutions were stirred at 28 °C for 2 h at a speed of 800 rpm. Subsequently, the mixtures were combined in equimolar ratios with predesigned single-stranded DNA sequences in TEM buffer (10 mM Tris, 1 mM EDTA, 20 mM $MgCl_2$, pH = 8.0). The resulting solution was heated at 95 °C for 5 min and then annealed at 4 °C for at least 0.5 h using a T series Multi-Block Thermal Cycler (LongGene, China) to stabilize the structures. Finally, the prepared DbTACs or HPK1-DbTACs were stored at 4 °C until further use.

## Fabrication of bis-DbTACs

For the formation of bis-DbTACs, the single-stranded DNA S1 (50 μM), S2 (50 μM), and S4 (50 μM) were mixed with CDK9 ligand (500 μM), CDK6 ligand (500 μM), and CRBN ligand (500 μM) at a 1:1 molar ratio, respectively. The three solutions were stirred at 28 °C for 2 h at a speed of 800 rpm. The subsequent steps are described as mentioned above. The DNA sequences used are shown in Supplementary Table 2.

## Fabrication of Abs-DbTACs

The DNA sequences used are shown in Supplementary Table 4. The preparation of antibody-DbTACs conjugates followed a previously reported protocol[41] with modifications.

Firstly, 1 eq. CDK9 polyclonal antibody was buffer-exchanged into 1× PBS buffer (pH 7.2) and treated with 80 eq. of the heterobifunctional N-succinimidyl 3-maleimidoproppionate (BMPS) linker (6.67 μL in DMSO) overnight at room temperature. The BMPS-modified antibody was then purified using a NAP-5 desalting column and concentrated by vacuum. Next, the 5′-thiol-modified DNA S1 strand was treated with tris (2-carboxyethyl) phosphine hydrochloride (TCEP)-containing TEM buffer (pH 8.0) for 2.5 h at room temperature. Separately, the DNA S4 strand containing the DBCO group and azide-modified pomalidomide was catalyzed to form S4–L2 by copper-free click chemistry. The above solutions and the other two strands forming DNA tetrahedra were mixed at a 1:1:1:1 eq. ratio to synthesize HS-tDNA through thermal annealing. The HS-tDNA pellet was then ethanol-precipitated to remove excess TCEP and dissolved with the solution containing the concentrated antibody-crosslinker directly. The mixture was incubated overnight at room temperature, and the constituents of the antibody-modified DbTACs mixture were analyzed by agarose gel electrophoresis (100 V for 30 min). The prepared Abs-DbTACs were stored at 4 °C for further use.

## Fabrication of Oligo-DbTACs

The DNA sequences used are shown in Supplementary Table 4. To synthesize Oligo-DbTACs, the DNA S4 strand containing a DBCO group and azide-modified pomalidomide was firstly catalyzed to form S4–L2 through copper-free click chemistry. Next, the S4–L2 strand was mixed with DNA S1_ERG(R), ERG(F), S2, and S3 chains at an equivalent ratio and thermally annealed to form Oligo-DbTACs. Similarly, a negative control was prepared using the same method but with the DNA S1_NC(R) and NC(F) strands. The resulting Oligo-DbTACs were stored at 4 °C until further use.

## Mass spectrometry analysis

The S1–L1 and S4–L2 samples were obtained with three replicates and analyzed by Sangon Biotech Co., Ltd. using ESI ion-hydrazine mass spectrometry (Thermo LTQ). A 30 μL sample injection at a final concentration of 1–5 μM was performed. Data analysis was conducted using Origin software (Version 2018, OriginLab Software Inc., USA).

## Gel separation of DbTACs, bis-DbTACs, Abs-DbTACs, and Oligo-DbTACs

Agarose gel (2 and 3%) was used for gel separation. In brief, 5 μL each sample was mixed with 1 μL of DNA loading buffer (6X) and loaded onto the gel. Electrophoresis separation was performed using a Bio-Rad electrophoresis system (USA) at 120 V for 20 min. The gels were visualized using a gel electrophoresis imaging system (Bio-Rad, USA).

For the verification of ligand covalent and self-assembly mechanism, PAGE (10%) was employed. Five microliters of each sample were mixed with 1 μL of DNA loading buffer (6×) and loaded onto the gel. Electrophoresis was conducted in 1× TBE buffer at 80 V for 80 min. The resultant gels were stained with SYBR green (10,000×, Solarbio Science & Technology Co., Ltd. (Beijing, China)) for 0.5 h in the dark and imaged using a Bio-Rad imaging system.

## Preparation and enrichment of Au NPs

Synthesis of Au NPs with different sizes (5, 10, and 15 nm) was carried out via the sodium citrate reduction method. In brief, 1% trisodium citrate (w/v, 4 mL, 4 mL, 4 mL), 1% tannic acid (w/v, 0.7 mL, 0.1 mL, 0.01 mL), and 0.1 M $K_2CO_3$ (0.2 mL, 0.025 mL, 0.0025 mL) were mixed, and then the mixed solution was diluted to 20 mL with deionized water and named solution 1. Then, 1% $HAuCl_4$ (w/v, 1 mL, 1 mL, 1 mL) was added to deionized water to obtain solution 2, which was mixed with solution 1 at a ratio of 1:4. The resulting mixture was immediately heated and stirred at 60 °C for 10 min until the color of the solution turned bright red. The obtained Au NPs were stored at 4 °C and characterized by TEM and particle size analysis.

Enriching Au NPs was performed according to guidelines. The prepared Au NPs were concentrated up to 100-fold with bis(p-sulfonatophenyl) phenylphosphine dihydrate dipotassium salt (BSPP) following established protocols[34,35,42]. The resulting BSPP-protected Au NPs were quantified using a UV–visible spectrophotometer.

## Imaging of DbTACs and bis-DbTACs equivalents by TEM

The DNA sequences used are shown in Supplementary Table 3. Firstly, BSPP-protected Au NPs (5, 10, and 15 nm) were mixed with single-stranded DNA to replace the ligand covalent sites with polyA domains for 1 min at room temperature. Citrate-HCl buffer was added to the mixture to a final concentration of 10 mM and incubated for 30 min. The resulting DNA-encoded BSPP-Au NPs were then mixed with other single-stranded DNA sequences of predesigned DbTACs or bis-DbTACs equivalents, heated at 95 °C for 5 min in the T series Multi-Block Thermal Cycler and annealed at 4 °C for at least 0.5 h to obtain stabilized DbTACs or bis-DbTAC equivalents.

To observe the generated DbTACs or bis-DbTACs equivalents, they were dropped onto a dry copper grid (Alied High Tech Products, Inc., Beijing, China) for 2 min. The samples were then observed and photographed using TEM (JEM-2000 EX II (JEOL Company, USA)).

## Stability of DbTACs and Abs-DbTACs

Specifically, DbTACs-26 Å (11.36 μM) was incubated with PBS and 10% FBS-contained IMDM medium at 37 °C for different time points, including 0, 1, 2, 4, 6, 12, and 24 h. At each time point, samples were taken and subjected to detection using 2% agarose gel electrophoresis to analyze the stability of the DbTACs.

Abs-DbTACs (7.41 μM) were incubated with PBS, 10% FBS-contained 1640 medium, protease (150 U/mL, Shanghai yuanye Bio-Technology Co., Ltd., S10051), or DNase I (150 U/mL, Shanghai yuanye Bio-Technology Co., Ltd., S10073) at 1:1 (v/v) at 37 °C for different time points, including 0, 1, 2, 4, 6, 12, and 24 h. The subsequent procedures were performed as previously described.

## UV–visible spectroscopy of Abs-DbTACs

The purified Abs-crosslinker was mixed with Bradford reagent (Sangon Biotech Co., Ltd.) at a 1:1 ratio, and the absorbance was measured by a NANODROP ONE instrument. In the presence of the antibody, a shift in the maximum absorption peak from 465 to 595 nm was observed.

## Cell culture

The acute myeloid leukemia cancer cell lines MV4–11 (catalog number CRL-9591), human hepatoma cells HepG2 (catalog number HB-8065) and human prostate cancer cells PC3 (catalog number CRL-1435), and human embryonic kidney (HEK) 293 T (catalog number CRL-3216) were purchased from American Type Culture Collection (ATCC, USA). Human T lymphocyte cell line Jurkat, Clone E6-1 (catalog number, TCHU123) was from Cell Bank/Stem Cell Bank, Chinese Academy of Sciences. The acute myeloid leukemia cancer cell line MOLM13 (catalog number iCell-h423) was purchased from iCell Bioscience Inc., Shanghai. MV4–11 cells were cultured in Iscove's modified Dulbecco's medium (IMDM, Invitrogen, Shanghai, China) supplemented with 10% FBS (Gibco) at 37 °C in a humidified atmosphere containing 5% $CO_2$. HepG2, HEK293T, and PC3 cells were incubated in Dulbecco's modified Eagle's medium (DMEM, Invitrogen, Shanghai, China) supplemented with 10% FBS at 37 °C in a humidified atmosphere containing 5% $CO_2$. MOLM13 and Jurkat cells were cultured in Roswell Park Memorial Institute (RPMI-1640, Invitrogen, Shanghai, China) supplemented with 10% FBS at 37 °C in a humidified atmosphere containing 5% $CO_2$.

**Subcellular localization of DbTACs.** HepG2 cells were seeded onto confocal dishes and incubated overnight for adhesion. The cells were then treated with Cy3-labeled DbTACs-26 Å (DbTACs-26 Å-Cy3) at a final concentration of 200 nM for various durations (0, 10 min, 3 h, 6 h, 12 h, and 24 h). After exposure to DbTACs-26 Å-Cy3, the cells were rinsed three times with PBS. To fix the cells, 4% paraformaldehyde was applied for 20 min, followed by PBS wash. Subsequently, 100 μL of DAPI dye (Jiangsu KeyGEN Bio TECH Corp., Ltd., KGA215-50) was added to the cells and allowed to stain at room temperature for 10 min, protected from light. The staining solution was then discarded, and the cells were rinsed twice with PBS. Finally, PBS was added, and inverted fluorescence microscopy (Leica DMi8, Germany) was employed to observe and capture images.

**Western blotting.** MV4–11 cells were seeded at a density of $2 \times 10^6$ cells/well in 6-well plates and allowed to adhere and grow overnight. Different concentrations (final concentrations of 80 and 200 nM) of DbTACs solutions (DbTACs-8 Å, −11 Å, −16 Å, −21 Å, −26 Å, and −57 Å) were added to the cells, along with positive control (B11), and negative controls (Control, tDNA). The cells were then co-incubated for 6 h at 37 °C in an incubator. After the incubation period, the cells were collected and subjected to WB analysis.

Jurkat cells were seeded at a density of $2 \times 10^6$ cells/well in 6-well plates and allowed to adhere and grow overnight. Subsequently, various concentrations (final concentrations of 50, 100, 200, 400, 800, and 1600 nM) of HPKE-DbTACs solutions were added to the cells, accompanied by negative controls at a final concentration of 1600 nM (Control, Sunitinib, tDNA, tDNA-L2, and tDNA-L4). The cells were co-incubated at 37 °C in an incubator for 12 h. The following steps are described above.

MV4-11, MOML13, PC3, or Jurkat cells were cultured in 6-well plates and treated with specific concentrations of compounds for specific times at 37 °C with 5% $CO_2$. The cells were collected by centrifugation, washed with ice-cold PBS three times, and lysed with RIPA lysis buffer (RIPA:PMSF = 99:1) (Hangzhou Fude Biological Technology) for 50 min on ice. The lysates were centrifuged at 14,847$g$ for 25 min at 4 °C. The protein concentration of the supernatants was measured using an enhanced BCA protein assay kit (Beyotime Biotechnology (Shanghai, China)). The protein extract was mixed with dual-color protein loading buffer, denatured in a 100 °C bath for 10 min, and loaded onto an SDS–PAGE gel. The proteins were then transferred to a nitrocellulose membrane by electrophoresis. Primary antibodies used in this study were rabbit GAPDH polyclonal antibody (Proteintech Group, Rosemont, IL, USA, 10494-1-AP, 1:10000), rabbit CDK9 polyclonal antibody (Proteintech Group, Rosemont, IL, USA,

11705-1-AP, 1:1000), mouse CDK1/2 (AN21.2) monoclonal antibody (Santa Cruz Biotechnology, sc-53219, 1:250), mouse CDK6 antibody (Proteintech Group, Rosemont, IL, USA, 66278-1-Ig, 1:1000), rabbit ERG polyclonal antibody (Proteintech Group, Rosemont, IL, USA, 14356-1-AP, 1:1000), rabbit HPK1 polyclonal antibody (Proteintech Group, Rosemont, IL, USA, 23950-1-AP, 1:1000). The secondary antibodies used were HRP-conjugated recombinant rabbit anti-mouse IgG kappa light chain (Proteintech Group, Rosemont, IL, USA, SA00001-19, 1:5000) and HRP-conjugated Affinity Pure goat anti-rabbit IgG (H + L) (Proteintech Group, Rosemont, IL, USA, SA00001-2, 1:10000). Western blot images were captured using UVITEC Imaging Systems (Uvitec Ltd., UK). The protein levels were quantified by the gray values of bands in the resulting images using ImageJ.

## Immunofluorescence staining

HepG2 cells were seeded into 10 mm confocal dishes and incubated overnight. Various DbTACs-linker lengths or bis-DbTACs (200 nM) were applied for 6 h at 37 °C with 5% $CO_2$. Subsequently, cells were quickly washed thrice with ice-cold PBS and then fixed with 4% paraformaldehyde for 20 min. After fixation, cells were washed with PBS and permeabilized with 0.1% Triton X-100 at room temperature for 10 min. The samples were then incubated in a blocking solution for 1 h at room temperature, followed by incubation with primary antibody overnight at 4 °C. The primary antibody used was mouse CDK6 antibody (Proteintech Group, Rosemont, IL, USA, 66278-1-Ig, 1:100), rabbit CDK9 polyclonal antibody (Proteintech Group, Rosemont, IL, USA, 11705-1-AP, 1:100). After washing thrice with PBST (PBS with 0.1% Tween 20), the samples were incubated with the secondary antibody in the dark for 1.5 h at room temperature. The secondary antibodies used were coraLite594-conjugated donkey anti-mouse IgG(H + L) (Proteintech Group, Rosemont, IL, USA, SA00013-7, 1:100) and coraLite488-conjugated donkey anti-rabbit IgG(H + L) (Proteintech Group, Rosemont, IL, USA, SA00013-6, 1:100). The nuclei were counterstained with DAPI (Jiangsu KeyGEN Bio TECH Corp., Ltd, KGA215-50, 1 μg/mL, 100 μL) at room temperature for 10 min, followed by washing. Images were acquired using an inverted fluorescence microscope or a confocal laser microscope (FV1000, Olympus, Japan).

## Plasmid construction

The expression plasmid CDK9_pcDNA3.1(+)-N-eGFP was constructed by inserting a synthesized gene encoding CDK9 into the pcDNA3.1(+) vector. The vector contains an eGFP signal sequence, and the insertion was performed using the multiple cloning sites (MCS) BamHI/EcoRI.

## Live cell real-time imaging of CDK9 content

HEK293T cells were cultured into 10 mm confocal dishes until they reached 70% confluency and were then incubated overnight. The CDK9-eGFP plasmid was constructed and transfected into HEK293T cells using Lipofectamine™ 2000 transfection reagent (Thermo Fisher Scientific, USA) for coculture for 6 h. After 6 h, the medium was changed to FBS-containing DMEM. The expression of CDK9−eGFP fusion protein with green fluorescence was recorded by an inverted fluorescence microscope after 65 h. To investigate the dynamics of CDK9 protein content in living cells, changes in fluorescence intensity were further observed by administering DbTACs for 6 h and 12 h.

## Flow cytometry of CDK9 on live cells

HEK293T cells were seeded into 6-well plates and incubated overnight to reach approximately 70% confluency. After successfully expressing CDK9-eGFP fusion protein in HEK293T cells, the PBS, DbTACs-26 Å, and B11 (final concentrations of 200 nM) were then added to cells and incubated for an additional 12 h. After treatment, cells were detached from the wells using trypsin-EDTA, washed and resuspended with PBS at a concentration of $1 \times 10^6$ cells/mL. CDK9-eGFP levels were

measured using flow cytometry (BD Accuri® C6, USA) equipped with a 488 nm laser for excitation and a 530/30 nm bandpass filter for detection of GFP fluorescence. Data were analyzed using FlowJo software (BD Biosciences, USA).

## Cell apoptosis assay

The MV4–11 cells ($4.0 \times 10^5$ cells/well) were seeded into each well of a 6-well transparent plate and incubated overnight. After 24 h of seeding, the test compounds were added at final concentrations of 0, 80, 100, 150, 200, and 250 nM, and the cells were incubated for another 6 h. Next, the cells were collected by centrifugation at 100g for 5 min and washed twice with PBS. The staining solution containing annexin V-FITC and propidium iodide (PI) was prepared according to the manufacturer's instructions (Beyotime Biotechnology, Shanghai, China). Then, the cells were resuspended in 500 µL of binding buffer containing the staining solution and incubated at room temperature for 20 min away from light. Finally, the samples were analyzed by flow cytometry in an ice bath, and the data were analyzed using FlowJo software.

## Cell autophagy assay

MV4–11 cells were seeded into a 96-well blackboard and allowed to adhere overnight. Subsequently, the cells were treated with various compounds for a duration of 12 h. As a positive control, an autophagy inducer (Beyotime Biotechnology, Earle's Balanced Salt Solution) was applied for 4 h. After treatment, the cells were stained using the Autophagy Staining Assay Kit with MDC (Beyotime Biotechnology, C3018S) following the provided instructions. The fluorescence microplate (SpectraMax® iD3) was employed to evaluate the levels of autophagy in MV4–11 cells post-treatment. The optical density (OD) values were measured at an excitation wavelength of 335 nm and an emission wavelength of 512 nm.

## Cell cycle assay

Briefly, MOLM13 cells were seeded in 6-well plates and treated with bis-DbTACs (final concentrations of 80, 200, 250, 500, and 1000 nM) or PBS (Control) for 12 h. After the treatment period, cells were harvested and fixed in ethanol. Fixed cells were then stained using the Cell Cycle Detection Kit (Jiangsu KeyGEN Bio TECH Corp., Ltd., KGA, KGA512) following the manufacturer's protocol. The flow cytometry data were processed using ImageJ to determine the distribution of cells in different phases of the cell cycle, including the G1, S, and G2 phases.

## qRT–PCR analysis

The MV4–11 cells were seeded into 6-well plates and allowed to incubate overnight. DbTACs-26 Å (final concentrations of 0, 80, and 200 nM) was incubated with cells for 6 h, then cells were collected and RNA was extracted using the TRIzol method. The RNA purity and concentration were measured using a NANODROP ONE spectrophotometer. The cDNA was synthesized from the extracted RNA using the PrimeScript™ RT reagent kit (Takara Bio, Japan). The primer sequences used for qRT-PCR analysis were as follows: CDK9_F: 5′-CTCTGCGGCTCCATCAC-3′, CDK9_R: 5′-GCCTGTCCTTCACCTTCC-3′; MCL-1_F: 5′-GATGTGAAATCGTTGTCTCGAG-3′, MCL-1_R: 5′-GAAAT-GAGAGTCACAATCCTGC-3′; Actin_F: 5′-CCTCACTGTCCACCTTCC-3′, Actin_R: 5′-GGGTGTAAAACGCAGCTC-3′. qRT-PCR was performed on a Real-Time PCR Instrument (Thermo Fisher Scientific, USA) using the following protocol: initial incubation at 37 °C for 15 min, followed by 5 s at 85 °C to inactivate previous amplicons with uracil-DNA glycosylase and a 2 min incubation at 95 °C to activate the Taq polymerase. The amplification cycle, consisting of 15 s at 95 °C, 30 s at 60 °C, and 30 s at 70 °C, was repeated 40 times. The relative expression levels of *CDK9* and *MCL-1* were determined using the ΔΔCt method (Ct gene of interest−Ct internal control). Actin was used as the internal control.

The results were presented as the fold change relative to the control group and plotted. All experiments were repeated six times.

## Quantitative proteomics analysis

The 4D-FastDIA-based quantitative proteomic analysis of human MV4–11 cells was carried out by Jingjie PTM Biolabs Inc. (Hangzhou, China). Samples were sonicated three times on ice using a high-intensity ultrasonic processor (Scientz) in lysis buffer (8 M urea, 1% protease inhibitor cocktail). The remaining debris was removed by centrifugation at 12,000g at 4 °C for 10 min. Finally, the supernatant was collected, and the protein concentration was determined with a BCA kit according to the manufacturer's instructions.

For digestion, the protein solution was reduced with 5 mM dithiothreitol for 30 min at 56 °C and alkylated with 11 mM iodoacetamide for 15 min at room temperature in darkness. The protein sample was then diluted by adding 100 mM TEAB to urea concentration less than 2 M. Subsequently, trypsin was added at 1:50 trypsin-to-protein mass ratio for the first digestion overnight and 1:100 trypsin-to-protein mass ratio for a second 4 h-digestion. Finally, the peptides were desalted by the C18 SPE column.

The tryptic peptides were dissolved in solvent A (0.1% formic acid, 2% acetonitrile/in water) and directly loaded onto a homemade reversed-phase analytical column (25 cm length, 75/100 µm i.d.). Peptides were separated with a gradient from 6 to 24% solvent B (0.1% formic acid in acetonitrile) over 70 min, 24–35% in 14 min, and climbing to 80% in 3 min then holding at 80% for the last 3 min, all at a constant flow rate of 450 nL/min on a nanoElute UHPLC system (Bruker Daltonics).

The peptides were subjected to a capillary source followed by the timsTOF Pro (Bruker Daltonics) mass spectrometry. The electrospray voltage applied was 1.60 kV. Precursors and fragments were analyzed at the TOF detector, with an MS/MS scan range from 100 to 1700 m/z. The timsTOF Pro was operated in parallel accumulation serial fragmentation (PASEF) mode. Precursors with charge states 0–5 were selected for fragmentation, and 10 PASEF-MS/MS scans were acquired per cycle. The dynamic exclusion was set to 30 s.

The resulting MS/MS data were processed using MaxQuant search engine (v.1.6.15.0). Tandem mass spectra were searched against the human SwissProt database (20,422 entries) concatenated with the reverse decoy database. Trypsin/P was specified as a cleavage enzyme allowing up to two missing cleavages. The mass tolerance for precursor ions was set as 20 ppm in the first search, and 5 ppm in the main search, and the mass tolerance for fragment ions was set as 0.02 Da. Carbamidomethyl on Cys was specified as a fixed modification, and acetylation on protein N-terminal and oxidation on Met were specified as variable modifications. FDR was adjusted to <1%.

Then, PCA was used to evaluate the repeatability of samples from each group. Differential proteins were obtained after the qualification of samples, whose differences in relative quantification in two groups were compared by *t*-test, and the corresponding *p*-value was calculated. In addition, with a criterion of *p*-value ≤ 0.05, the protein ratio > 1.5 was regarded as upregulation, while the protein ratio < 1/1.5 as downregulation.

Based on the identified proteins, the subcellular localization analysis was performed using the WoLF-PSORT database; GO annotation is to annotate and analyze the identified proteins with eggnog-mapper software (v2.1.6). The software is based on the EggNOG database (v5.0.2, http://eggnog5.embl.de/#/app/home). Extracting the GO ID from the results of each protein note and then classifying the protein according to Cellular Component, Molecular Function, and Biological Process; Kyoto Encyclopedia of Genes and Genomes (KEGG) database (v5.0, http://www.kegg.jp/kegg/mapper.html) was used for KEGG pathway enrichment analysis. Fisher's exact test was used to analyze the significance of KEGG pathway enrichment of differentially expressed proteins (using the identified protein as the background),

and $P$ value $< 0.05$ were considered significant. Furthermore, all differentially expressed protein database accession or sequence were searched against the STRING database (v11.5, https://cn.string-db.org/) for protein–protein interactions. Only interactions between the proteins belonging to the searched data set were selected, thereby excluding external candidates. STRING defines a metric called "confidence score" to define protein-protein interaction (PPI) confidence; we fetched all interactions that had a confidence score $\geq 0.7$ (high confidence). PPI network form STRING was visualized using the R package "networkD3" tool.

### SEL-HPLC analysis of the formation of ternary complexes
DbTACs-26 Å were preincubated with either human recombinant CDK9 (Solarbio Science & Technology Co., Ltd. (Beijing, China)), CRBN (Cloud-Clone Corp (Wuhan, China)) or both proteins together at room temperature for 0.5 h in an equal molar ratio. The formation of ternary complexes was analyzed by size exclusion high-performance liquid chromatography (SEC-HPLC) using an FLM Scientific Instrument Co., Ltd. (Guangzhou, China) column, with the retention time of the samples being measured at 260 nm. The mobile phase used was Tris-NaCl buffer with a pH of 7.4, and a flow rate of 0.8 mL/min was maintained throughout the analysis.

### SEL-HPLC analysis of Abs-DbTACs
Abs-DbTACs, HS-tDNA, and free CDK9 polyclonal antibody (final concentrations of 200 nM) were analyzed by an SEC-HPLC column (FLM Scientific Instrument Co., Ltd. (Guangzhou, China)). PBS was used as mobile phase, pH = 7.4, and flow rate: 0.8 mL/min. The retention time of the samples was monitored and analyzed to determine the composition of the complexes.

### All-atom models of DbTACs
All-atom models of various DbTACs, bis-DbTACs, Abs-DbTACs, Oligo-DbTACs, etc., were constructed using the PolygenDNA program[43,44] and MOE software. To generate an all-atom model of a DNA tetrahedron using the PolygenDNA program, the DNA sequence of interest was specified as input. A double-stranded DNA helix was then generated, and the tetrahedral vertices were placed at the desired positions in 3D space. A series of energy minimization and geometry optimization steps were applied to refine the initial atom positions and adjust the geometry of the helix to achieve optimal bond lengths, bond angles, and dihedral angles. The resulting pdb file contained all-atom coordinates for all atoms in the DNA tetrahedron, including hydrogen atoms and other small molecules. Molecular visualization software, such as MOE, was used to visualize and analyze the model and make any necessary adjustments. To generate the final pdb file of series DbTACs by covalently linking the DNA tetrahedra and ligands, MOE was used. Firstly, the 3D coordinates of the ligands were generated and optimized using the MOE Builder module. Next, the DNA tetrahedra pdb file was loaded into MOE, and the 3D coordinates of the DNA tetrahedra were optimized using the MOE Protein Preparation Wizard. The ligands were then docked into the optimized DNA tetrahedra structure using the MOE Dock module, and the covalent bonds between the ligands and DNA tetrahedra were formed using the MOE Editor module. Finally, the resulting structure was energy minimized using the MOE Energy Minimization module to obtain the all-atom model of various DbTACs. MOE employs various algorithms to minimize the energy of the structure and optimize the geometry of the molecule. The optimized pdb file of various DbTACs was used for subsequent analysis and simulations.

### Molecular docking analysis
The molecular docking studies of DbTACs-linker lengths with CDK9 protein and DbTACs-linker lengths with CDK9-CRBN proteins were performed using the MOE software package developed by Chemical Computing Group. The crystal structures of CDK9 protein (pdb ID: 3BLH) and CRBN protein (pdb ID: 4CI3) were downloaded from the Protein Data Bank (PDB) and prepared using the MOE Protein Preparation Wizard.

For the DbTACs-linker lengths with CDK9 docking, the DbTACs pdb file generated from the previous step was loaded into MOE. Prior to docking, the protein underwent preprocessing using QuickPrep with default options, which included rectifying structural inaccuracies, incorporating hydrogen atoms, optimizing three-dimensional H-bonding networks, eliminating water molecules beyond 4.5 Å from the protein, and minimizing within a limited range of 8 Å of the altered base pairs. Next, the CDK9 ligand on the DbTACs was identified as the Ligand Site using the MOE Site Finder module, and a docking box was defined around the site. The native ligand pockets of the CDK9 protein were selected as the Receptor Site in the docking studies. The MOE Dock module was then used to dock DbTACs into the binding site of CDK9 using a Refinement with Rigid Body protocol, which generated several docking poses based on the predicted interaction energies between the ligand and receptor. The docking poses were ranked according to their binding affinities, and their Docking Score S[45] was recorded. The pose with the highest Docking Score was selected as the final result. The Docking score S was calculated based on the following formula 1:

$$\Delta G_{\mathrm{Binding}}^{\mathrm{Calc.}} = \alpha \left( \underbrace{\frac{2}{3}\left(E_{\mathrm{Inter}}^{\mathrm{Coul}}\right) + \Delta G_{\mathrm{Bind}}^{\mathrm{R}}}_{\Delta G_{\mathrm{Bind}}^{\mathrm{Elec}}} + \underbrace{E_{\mathrm{Inter}}^{\mathrm{vdW}} + \Delta G_{\mathrm{Bind}}^{\mathrm{npsol}}}_{\Delta G_{\mathrm{Bind}}^{\mathrm{Non-polar}}} \right) + c \qquad (1)$$

The Docking scoring S was primarily composed of $\Delta G_{\mathrm{Binding}}^{\mathrm{Calc.}}$, which included $\Delta G_{\mathrm{Bind}}^{\mathrm{Elec.}}$ and $\Delta G_{\mathrm{Bind}}^{\mathrm{Non-polar}}$. $E_{\mathrm{Inter}}^{\mathrm{Coul.}}$ and $E_{\mathrm{Inter}}^{\mathrm{vdW}}$ represented the columbic and van der Waals contribution to binding, respectively. $\Delta G_{\mathrm{Bind}}^{\mathrm{R}}$ was the change in reaction field energy upon binding. The $\Delta G_{\mathrm{Bind}}^{\mathrm{npsol}}$ term represented the change in non-polar solvation (van der Waals and cavitation cost) upon binding. Furthermore, the scaling factor for electrostatic interactions was empirically determined to be 2/3, which improved accuracy compared to the theoretically ideal value of 1/2[46].

### SPR binding assay
For binary binding experiments, stock solutions of DbTACs-26 Å or free CRBN protein were serially diluted in PBS-P (containing 0.5% Surfactant P20) running buffer (twofold serial dilution). The diluted solutions were injected over a CM5 chip coated with immobilized CDK9.

For the ternary binding study, DbTACs-8 Å/DbTACs-26 Å/DbTACs-57 Å were mixed with a solution of CRBN protein to prepare a final solution of 400 nM DbTACs and 800 nM CRBN protein in PBS-P running buffer. The complexes were preincubated in PBS-P running buffer for 0.5 h, followed by serial dilutions (six-point twofold serial dilutions).

SPR binding responses for binary and ternary complexes were performed in multicycle kinetic mode at 298.15 K with a contact time of 60 s, a flow rate of 30 μL min$^{-1}$, and a dissociation time of 60 s. The raw sensorgram data was processed using Biacore T200 Evaluation Software. The reference surface and blank injections were subtracted from the raw data before data analysis. The steady-state affinity (SSA) model was used to calculate the association rate ($k_{\mathrm{on}}$), dissociation rate ($k_{\mathrm{off}}$), and dissociation constant ($K_{\mathrm{D}}$) for the binding affinity between binary and ternary complexes.

### Statistics and reproducibility
The results are presented as the mean ± standard deviation (SD). Differences between groups were analyzed for statistical significance using $t$-test analysis in GraphPad Prism (version 8, GraphPad Software Inc., USA). Statistical significance was

accepted with $P < 0.05$, *: $P < 0.05$, **: $P < 0.01$, ***: $P < 0.001$, ****: $P < 0.0001$, and n.s.: no significant difference. Each experiment was independently repeated three times, and similar results were obtained. The data analysis was conducted by using Origin software (Version 2018, OriginLab Software Inc., USA).

## Reporting summary

Further information on research design is available in the Nature Portfolio Reporting Summary linked to this article.

## Data availability

All data generated in this study are provided in the Supplementary Information/Source Data file. The crystal structures of CDK9 protein (pdb ID: 3BLH) and CRBN protein (pdb ID: 4CI3) were downloaded from the Protein Data Bank (PDB). The proteomics data used in this study are available in the PRIDE database under accession code PXD042665. Source data are provided in this paper.

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

## Acknowledgements

We gratefully acknowledge financial support from the National Natural Science Foundation of China (82072058 to Y.M., 82230070 to Y.Q.G., 82073702 to J.L.B.), and Outstanding Youth Foundation of Jiangsu Province of China (CN) (BK20220150 to Y.M. and BK20211580 to J.L.B.), Qinglan Project of Jiangsu Province of China, the National Innovation and Entrepreneurship Training Program for Undergraduate (202110316015, 2021103160022Zs, and 202110316014Z to J.L.B.), "Double First-Class201D university project (CPUQNJC22_05 to J.L.B.).

## Author contributions

L.Z., J.B., and Y.M. conceived the project. B.Y. and R.C. synthesized, purified, and characterized ligands 1–4. L.Z. and M.G. carried out the visualization characterization experiments and interpreted data. L.Z., B.Y., and M.G. carried out and analyzed in vitro experiments. Z.L. and Y.G. oversaw and provided insights and materials. L.Z., J.B., and Y.M. wrote the paper with input from all authors. J.B. and Y.M. provided supervision.

## Competing interests

The authors declare no competing interests.
