## [Peer Review File · Nature Communications]

REVIEWER COMMENTS

Reviewer #1 (Remarks to the Author):

This paper by Zhou et al. describes the use of a DNA framework as a platform for targeted protein degradation. In their paper, the authors report the use of a tetrahedral DNA structure to display small molecules, antibodies or nucleic acid to induce the degradation of different proteins in cellulo. This report is the first example of the use of DNA as a linker for heterobifunctional molecules. By using the programmability of nucleic acids, the authors can display small molecules with great precision based on the number of nucleobases used as spacers between the tetrahedron and the ligand. CDK9 and ERG are used as model systems and the authors show degradation of both targets. The authors clearly demonstrated formation of a supramolecular organization (ternary or even quaternary complex formation). Overall, this strategy is very elegant and provides the advantage of being extremely modular and overcomes some of the limitation of PROTACs, e.g. the need to identify small molecules since antibodies or nucleic acids can be used directly. Despite the amount of data provided and the creativity shown by the authors, this paper needs to go through major revisions before being accepted into Nature Communications.

1. The syntax used throughout the paper makes it sometimes difficult to follow the authors train of thought. For the sake of clarity, the authors may want to revisit the structure and wording of some sentences. This will greatly improve the impact of the paper and convey their message to the reader in a more efficient fashion. For instance (among many others):

- Line 81: I am not sure what the authors mean by PROTAC tag.

- Line 110-113: consider revising the wording of the sentence.

- Line 117-118: none of the ligands used in this paper are covalent. Only the attachment to the DNA strand is.

2. In the paper, the authors focused their efforts on nuclear protein involved to some extent with cell cycle regulation or transcription as targets.

i. Does the DNA tetrahedron reach the nucleus after endocytosis?

ii. To support the author's claim that this technology is generalizable to other proteins, it would have been great to see proteins in other location being degraded. It seems like the DNA architecture plays a major role in the formation of a ternary complex and I question if this can happen for any protein or mostly for those involved in cell cycle or DNA transcription.

3. On a similar note, to the question above, since the linker relies on nucleic acids, is there any risk of off-target effects? Are DNA binding proteins degraded when DbTACs are used? Could the authors comment on the chances of this happening? Similarly, could this be an advantage, and could the authors exploit their technology to target DNA-binding proteins more efficiently to degradation?

4. Throughout the paper, one of the main controls missing is the use of the DNA tetrahedron alone, not labeled with any small molecule, protein or oligonucleotide. This control is key to demonstrate that the DNA architecture used is not responsible for some of the phenotypes and decrease in protein observed.

5. Figure 2 and Lines 195-196: The use of gold nanoparticles clearly shows the localization of the two modifications on the DNA architecture. I wonder if tagging the DNA architecture with fluorophore and demonstrating the cell uptake could be beneficial and inform on the intracellular localization of DbTACs.

6. Lines 200-210 and Figure 3.A) : In the quantification of the Western Blot, all conditions have a significant error bar except for 21 and 26 Angstrom DbTACs. Is this discrepancy due to a difference in sample number? Have the experiments been replicated more times with these DbTACs than the other ones? The authors also claim better degradation than the PROTAC B11, while the WB shows similar levels of degradation between B11 and 26A DbTAC.

7. In some cases, e.g. Figure 3A and Fig 3C, it looks like some parts of the blot are darker or lighter. It would be great to add the full Western Blot image in the supporting information to make sure the decrease observed is not an artifact.

8. Lines 272-276: The authors define cooperativity using KD_{ternary} twice instead of its real definition which is $\alpha = KD_{\text{ternary}} / KD_{\text{binary}}$.

9. Fig 4.d-g I am surprised by the low RU obtained in the SPR experiment. Based on the MW of the DbTACs, the theoretical maximum RU observed with 500 RU of CDK9 immobilized should be much higher. Is that small change due to a high reference binding from the DbTAC and the protein? On figure 4b, the change in retention time is also minimal despite the molecular weight almost doubling when both proteins are present.

10. Paragraph starting at line 370: Generalizability of DbTACs.

a. The authors re-use CDK9 as a target using an antibody. It would have been interesting to target a protein without a small molecule ligand in this case.

b. In this paragraph, the authors do not comment on the linker length used to display the CRBN ligand. Is 26A still preferable? It is expected that the use of an antibody or oligonucleotide would change the shape of the ternary complex induced. Were different linker lengths tested to achieve a good level of degradation?

c. Does the presence of the antibody or the oligonucleotide affect the cell permeability of the DbTACs? Is the half-life of the antibody conjugate also 6h or is degradation by proteases or nucleases a concern?

11. In their Discussion on line 453-455, the authors claim that their technology is a “one for all” strategy. I am not sure the small diversity of proteins used here allow them to make that claim. The questions above remain and I would expect ternary complexes to be very different depending on which type of target protein binder they use.

12. Line 461: DbTACs are without a doubt an impressive and exciting new technology. The authors achieved degradation of various targets intracellularly. However, I do not think they can claim that DbTACs can serve as a tool for SAR in linker type or length for the design of heterobifunctional molecules. Indeed, in their model of ternary complexes, different faces of the protein interact together. The side of the protein facing each other in PROTACs are facing the DNA tetrahedron in DbTACs and the target protein interacts with the E3 ligase on one side.

13. Line 473: It is not clear what the authors mean by the products require phase transfer. Please explain.

14. Lines 497-500: The authors claim that DbTACs can be synthesized in only two steps. However, this is only counting the steps necessary to form the DNA tetrahedron. The small molecule ligands, modified with reactive moieties still need to be synthesized. The gain of time is marginal compared to classical heterobifunctional molecules.

15. Methods: In general, the method section is not detailed enough. In too many cases are conditions missing to reproduce these experiments.

For instance, Line 688 Fabrication of DbTACs: in this section, the concentrations used in the click reaction are not described, only the stoichiometry is reported. Or Line 944 SPR Binding Assay: The

authors only reported running buffer without precising what this buffer is. Even if the buffer is commercial, it could be HEPES or PBS.

The authors need to improve the accuracy and the level of details of their methods to ensure that these results can be reproduced and the technology used in other labs.

16. Supplemental Information, Small molecule analysis: The analytical data for the small molecules synthesized is minimal. The authors provide ¹H NMR and HRMS, for a journal like Nature Communications, it would be good to provide ¹³C for each small molecule synthesized. In the HRMS spectrum, the X-axis is really zoomed in for two out of the three compounds reported. The authors should provide the full spectrum and explain what the smaller intensities peaks observed are. Overall, this level of characterization is insufficient. Additional HPLC traces could also improve the quality of the analysis.

Reviewer #2 (Remarks to the Author):

In general, the potential of PROTACs to modify targets (POIs) thought to be unreachable, such as because they lack well-defined binding sites, and to down-regulate all of the POI's activities, has sparked a great deal of interest in this approach. E3 ligase ligands for either von Hippel-Lindau (VHL) or Cereblon (CRBN) are present in the majority of PROTACs identified in the literature. All PROTACs, with the exception of one, that have started clinical trials are based on CRBN, and this makes CRBN-based PROTACs very interesting. In this paper, Li Zhou et al. have developed a DbTACs-based approach, where smart DNA frameworks are used to combine substrates, providing a single universal platform for tackling many challenges. This intriguing technique not only sheds light on generic degrader design principles, but it also offers up new avenues for drug development in precision medicine. Briefly, DNA tetrahedra were used as templates in this study, and the spatial locations of each atom were specified. Consequently, by accurately positioning POI and E3 ligase ligands on templates using the computer models, ligand spacings are controlled from 8 to 57 Å and the ideal linker length (26 Å) between ligands achieve a high degradation rate (70%) and higher binding affinity (K_D = 42 nM). Moreover, bispecific DbTACs (bis-DbTACs) accomplished multi-target depletion while preserving highly selective degradation of protein subtypes by combining trivalent ligands in a single DbTACs platform. This proof-of-concept study on DbTACs also applied to different types of warheads (for example, small molecules, antibodies, or DNA motifs). This reviewer is a

PROTAC computational design expert, so my comments will mostly address the simulation part. However, I must stress that, in spite of the drawbacks mentioned below, all sections are clearly written and understandable for the broad readership of Nature Communication. The subject of the paper is quite important, and the approach for the computational design of DbTACs is very interesting. However, in my opinion, the paper suffers from some critical issues (as detailed below) that prevent its publication at the current stage. Owing to the importance of the theme and the suitability of the approach, I believe the authors should be given a chance to perform a major revision on the manuscript addressing those critical points.

The details provided for the different POI and E3L complexes from the modeling study in the manuscript and supporting information are not clear, and it is hard to follow and reproduce the work reported in the manuscript, at least using the current description. Therefore, authors should add more description on how the POI and E3L were docked and further extend the validation procedure on ranking of docking poses using the different linker lengths. In several places in the manuscript, the author refers to computational models, including in lines 172-175 of Figure 8, but it was hard to find enough detail in the method section and in the supporting information. In order to reproduce or to re-create this model for different POI with appropriate linker length, computational detail is important.

Reviewer #3 (Remarks to the Author):

In this study, the authors have developed DNA framework-engineered chimeras platform to enable selectively targeted protein degradation, for which they named DbTACs. DNA tetrahedra were employed as templates for the synthesis of PROTAC degraders by precisely locating ligands of POI and E3 ligase on templates, ligand spacings were controllably manipulated from 8 Å to 57 Å. In one example, the authors showed that their designed DbTAC degrader is effective in reducing CDK9 protein by 70% when the linker length is 26 Å. Impressively, this degrader also shows selectivity over other CDK proteins. The authors further showed that multitargeted DbTAC degraders can be designed, using CDK6 and CDK9 as examples. The authors further demonstrated that ligands for POIs can be extended to antibody or DNAs using an antibody against CDK9 and DNAs targeting ERG. Overall, this is a very interesting study, suggesting that the DNA framework-engineered chimeras platform can be used as a general platform to target many proteins using different types of ligands. Hence, it is recommended that the paper be published in Nature Communications with the following revisions.

First, in the TPD field, proteomics has been widely used to demonstrate the selectivity of designed degraders. It is recommended that the authors perform a proteomics analysis at least on one representative degrader to demonstrate its selectivity more broadly.

Second, protein degradation certainly leads to phenotypes in cells. It is recommended that the authors show the phenotypes of protein degradation with at least some of those degraders shown in this study.

Point-by-point Response to Reviewer Comments

Reviewer 1

Comments:

This paper by Zhou et al describes the use of a DNA framework as a platform for targeted protein degradation. In their paper, the authors report the use of a tetrahedral DNA structure to display small molecules, antibodies or nucleic acid to induce the degradation of different proteins in cellulose. This report is the first example of the use of DNA as a linker for heterobifunctional molecules. By using the programmability of nucleic acids, the authors can display small molecules with great precision based on the number of nucleobases used as spacers between the tetrahedron and the ligand. CDK9 and ERG are used as model systems and the authors show degradation of both targets. The authors clearly demonstrated formation of a supramolecular organization (ternary or even quaternary complex formation). Overall, this strategy is very elegant and provides the advantage of being extremely modular and overcomes some of the limitation of PROTACs, e.g. the need to identify small molecules since antibodies or nucleic acids can be used directly. Despite the amount of data provided and the creativity shown by the authors, this paper needs to go through major revisions before being accepted into Nature Communications.

1. The syntax used throughout the paper makes it sometimes difficult to follow the authors train of thought. For the sake of clarity, the authors may want to revisit the structure and wording of some sentences. This will greatly improve the impact of the paper and convey their message to the reader in a more efficient fashion. For instance (among many others):

-Line 81: I am not sure what the authors mean by PROTAC tag.

-Line 110-113: consider revising the wording of the sentence.

-Line 117-118: none of the ligands used in this paper are covalent. Only the attachment to the DNA strand is.

Response: Thank you for making such valuable suggestions for improving the clarity and impact of our paper. To improve the quality and style of our writing, the whole manuscript has been polished by the language editing service at Springer Nature Author Services, without altering its original meaning. Please find attached our editorial certificate. And following is the corresponding revision of the above three instances.

Dear Li Zhou,

Thank you for choosing Springer Nature Author Services. This manuscript, titled "DNA framework-engineered chimeras enable selective targeted protein degradation," is very interesting. The paper was edited for grammar, phrasing, and punctuation. In addition, many edits were made to further improve the flow and readability of the text. Below, we highlight the areas of this paper that we focused on in our edit.

Some sentences were restructured to address overly complex structure or to revise potentially unclear phrasing.

Articles are an important aspect of the English language, including the definite article "the" and the indefinite articles "a" and "an." Our edits focused on improving article use, which is often strongly dependent on context and field conventions.

We made several edits to ensure natural and precise preposition usage, which is a difficult aspect of the English language. Prepositions include words such as "on," "for," "by" and "with."

Comments were left if further clarification would be helpful or confirmation of the meaning of the text was necessary. Please review these comments and all our changes carefully for more detailed suggestions, as well as to ensure that the final version of the manuscript is fully accurate.

Thank you again for using our editing services; we wish you the best of luck with your submission.

Best regards,

Laura M.
Senior Editor

-Line 81: I am not sure what the authors mean by PROTAC tag.

-We apologize for the lack of clarity in our language regarding the term "PROTAC tag" in the previous manuscript on line 81 (lines 78-81: "Inspired by the unique characteristics of DNA frameworks, the innovative strategy of DNA framework-based PROTACs (DbTACs) was developed by combining computational prediction and PROTACs tag technology"). The "PROTAC tag" refers to the heterobifunctional ligands present within the PROTAC structure, such as small molecules and antibodies. These ligands are responsible for specifically labeling the proteins of interest and the E3 ligases. To address the confusion, we have rephrased the sentence as follows: "Inspired by the unique characteristics of DNA frameworks, we have developed an innovative strategy for the development of DNA framework-based PROTACs (DbTACs) by combining computational prediction, DNA self-assembly, and PROTACs technology" in the Revised Manuscript Page 2 lines 80-84 with yellow background to better understand our manuscript.

-Line 110-113: consider revising the wording of the sentence.

-We appreciate your suggestion regarding the wording of the sentence in the previous manuscript on lines 110-113 (lines 110-113: "The active CDK9 ligand (L1) and CRBN ligand (L2) we studied earlier³² and modified with azide group here were engaged as the models"). To address the issue, we have revised the sentence in the Revised Manuscript Page 2 lines 113-120 with yellow background as follows: " In this study, we developed a novel strategy for the design and construction of DNA framework-based PROTACs (DbTACs) by combining computational prediction and click chemistry-mediated programmable linker technology. We utilized the active ligands (L1 and L2) of CDK9 and CRBN that were previously investigated³², which were subsequently modified with an azide group, to serve as our models". We hope this revision addresses your concerns and improves the clarity of our manuscript.

-Line 117-118: none of the ligands used in this paper are covalent. Only the attachment to the DNA strand is.

-We apologize for the confusion about our description in the previous manuscript lines 117-118 (lines 117-118: "DbTACs were designed and constructed with computer aid to completely expose covalent ligands outside the DNA tetrahedra"). Regarding your comment on lines 117-118, in fact, we confirm that all ligands used in our study were indeed covalently linked to DNA tetrahedra. We employed various covalent coupling methods (Response Fig. 1), including azide and DBCO group-mediated click chemistry for the synthesis of small molecule ligands-based DbTACs and bis-DbTACs, as illustrated in Fig. 2a and Fig. 6a in the revised manuscript. Additionally, we employed Michael addition for the synthesis of Abs-DbTACs, as depicted in Fig. 7a, and esterification reaction for Oligo-DbTACs, as shown in Fig. 8a.

To reflect these facts, we have revised the descriptions in the Revised Manuscript Page 2 lines 128-132, Page 8 lines 435-438, Page 9 lines 487-490, and Page 10 lines 542-547 with yellow background. Compared with physical attachment, these covalent couplings are essential for the stability and uniqueness of the DbTACs structure. The above descriptions were attached as follows for your convenience.

After revision (lines 128-132): Herein, dibenzocyclooctyne (DBCO)-modified DNA strand S1 coupled with azide-modified L1 through click chemistry (Fig. 2a(i)), which was selected as a fixed design. Meanwhile, the azide-modified L2 sites on DBCO-modified S4 sequences were reasonably

changed (Fig. 2a(iii)).

After revision (lines 435-438): Herein, a covalent ligand 3 (L3) was designed to target CDK6 protein and shared one E3 ligase ligand (L2) with CDK9 ligand (L1) on the DNA tetrahedral template (Fig. 6a).

After revision (lines 487-490): Firstly, CDK9 Abs and L2 were selected and conjugated onto DNA tetrahedra to prepare Abs-based DbTACs (Abs-DbTACs), further providing the universality of the platform.

After revision (lines 542-547): We designed and synthesized oligonucleotide-based DbTACs (Oligo-DbTACs) by introducing DNA tetrahedra as a linker conjugating L2 and ERG ligand, with binding moiety (purple) and three extra nucleotides (black) for protection of oligo degradation, according to literature modification³⁰ (Fig. 8a).

Response Fig. 1. Schematic diagrams of various covalent coupling methods used in this study. Various covalent coupling methods, including azide and DBCO group-mediated click chemistry (Fig. 2a, Fig. 6a), Michael addition (Fig. 7a), and esterification reaction (Fig. 8a).

2. In the paper, the authors focused their efforts on nuclear protein involved to some extent with cell cycle regulation or transcription as targets.

i. Does the DNA tetrahedron reach the nucleus after endocytosis?

ii. To support the author's claim that this technology is generalizable to other proteins, it would have been great to see proteins in other location being degraded. It seems like the DNA architecture plays a major role in the formation of a ternary complex and I question if this can happen for any protein or mostly for those involved in cell cycle or DNA transcription.

Response: Thank you for your valuable feedback. We appreciate your insightful comments and suggestions. We would like to provide the following response:

i. Several studies have reported that the average size of nuclear pores is about 40 ~ 100 nm [*Molecular biology of the cell*, 2002, 13, 425-434; *Nature reviews Molecular cell biology*, 2010, 11, 490-501; *Science*, 2006, 314, 815-817]. Small molecules smaller than 50 nm, such as ions, small proteins, and nucleic acid bases, can diffuse freely through the nuclear pore. The DNA tetrahedra used in our study were theoretically calculated to have a size of approximately 7.14 nm, indicating the possibility of nuclear localization.

Moreover, previous research by Chunhai Fan's team has investigated the cell entry and transport pathways of DNA tetrahedra [*Angew Chem Int Ed Engl*, 2014, 53, 7745-7750]. They demonstrated

that DNA tetrahedra were internalized via a caveolin-dependent pathway, transported to lysosomes, escaped from the lysosomes, and entered the cellular nuclei. In our study, we found that DbTACs could effectively degrade nuclear proteins based on the WB results, immunofluorescence data, etc, suggesting their presence in the nucleus and their functional impact.

Additionally, we conducted subcellular localization assays and observed the uptake of representative DbTACs-26 Å by HepG2 cells through endocytosis, followed by their entry into the nucleus (Supplementary Fig. 6 in the Revised Supplementary information). Thus, DbTACs do reach the nucleus after endocytosis. We have added corresponding findings and discussion to the Revised Manuscript Page 4 lines 206-214 and Revised Supplementary information Page 5 lines 43-48 with yellow background. A detailed method was added in Revised Manuscript Page 12 lines 825-839 with yellow background.

Please note that the changes have been marked in yellow in the revision:

After revision (lines 206-214): To investigate the fate of DbTACs after internalization, we conducted a subcellular localization assay using Cy3-labeled DbTACs-26 Å (DbTACs-26 Å-Cy3) in HepG2 cells. The cells were treated with DbTACs-26 Å-Cy3 for varying durations and subsequently stained with DAPI to track the intracellular localization of the compound. The results indicated that DbTACs-26 Å was effectively internalized by the cells and subsequently translocated to the nucleus (Supplementary Fig. 6).

After revision (lines 825-839): HepG2 cells were seeded onto confocal dishes and incubated overnight for adhesion. The cells were then treated with Cy3-labeled DbTACs-26 Å (DbTACs-26 Å-Cy3) at a final concentration of 200 nM for various durations (0, 10 min, 3 h, 6 h, 12 h, and 24 h). After exposure to DbTACs-26 Å-Cy3, the cells were rinsed three times with PBS. To fix the cells, 4% paraformaldehyde was applied for 20 min, followed by PBS wash. Subsequently, 100 µL of DAPI dye (Jiangsu KeyGEN Bio TECH Corp., Ltd, KGA215-50) was added to the cells and allowed to stain at room temperature for 10 min, protected from light. The staining solution was then discarded, and the cells were rinsed twice with PBS. Finally, PBS was added, and inverted fluorescence microscopy was employed to observe and capture images.

Supplementary Fig. 6. Subcellular localization of DbTACs-26 Å in HepG2 cells. HepG2 cells were treated with DbTACs-26 Å-Cy3 for different durations (0, 10 min, 3 h, 6 h, 12 h, and 24 h) and subsequently stained with DAPI. The red fluorescence represents the labeling of DbTACs-26 Å, while the blue color indicates the cell nucleus stained with DAPI. The Merge layer in the image displays the colocalization of DbTACs-26 Å, indicated by a white arrow. Scale bar: 20 µm.

ii. We appreciate the reviewer's suggestion that it would be interesting to explore the generality of our approach to proteins located in different cellular compartments. As our previous study focused on CDK family proteins, which are localized in the nucleus and associated with the cell cycle or DNA transcription, we aimed to demonstrate the universality of our approach by selecting hematopoietic progenitor kinase 1 (HPK1) as the target protein. HPK1 is primarily localized in the cytoplasm and is independent of the cell cycle or DNA transcription. It is an immunosuppressive regulatory kinase with a restricted expression profile in the hematopoietic compartment. Previous studies [*Cancer Cell*, 2020, 38, 551-566; *J Biol Chem*, 2012, 287, 34091-34100; *Nat Immunol*, 2007, 8, 84-91] have suggested that cytoplasmic HPK1 is a promising candidate target for cancer immunotherapies. To address this, we designed and synthesized novel HPK1-DbTACs, which co-conjugated the azide-modified HPK1 ligand (L4) and CRBN ligand (L2) on the DNA tetrahedral scaffold. Supplementary Western blotting (WB) results demonstrated that these HPK1-DbTACs effectively degraded HPK1 in a concentration-dependent manner compared to sunitinib (an HPK1 kinase inhibitor without a degradation effect) and negative controls (Control, tDNA, tDNA-L2, and tDNA-L4) (Supplementary Fig. 25a, b). The DC_{50} fit of HPK1 degradation was determined to be 262.0 nM (Supplementary Fig. 25c). Therefore, our results demonstrated successful targeted degradation of cytoplasmic proteins such as HPK1, in addition to nuclear proteins such as CDK6 and CDK9. These results and the corresponding discussion have been added to the Revised Manuscript Page 10 lines 565-581 and Revised Supplementary information Page 15 lines 137-147 with yellow background. Detailed methods were in the Revised Manuscript Page 11 lines 672-688 with yellow background. A detailed synthesis of azide-modified HPK1 ligand (L4) was also added in the Revised Supplementary information Page 23-24 lines 258-273 with yellow background. The above supplementary data, discussion and method were attached as follows for your convenience.

These findings validated the generality and versatility of our DbTACs strategy in different cellular compartments. These supplementary experiments also further verified that the degradation capability of DbTACs extends to proteins beyond those involved in cyclin-dependent kinase and DNA transcription processes.

Supplementary Fig. 25. Degradation of cytoplasmic proteins by DbTACs. a) WB analysis of the degradation of cytoplasmic HPK1 target in Jurkat cells by HPK1-DbTACs (final concentrations of 50, 100, 200, 400, 800, and 1600 nM) and negative controls (Control, Sunitinib, tDNA, tDNA-L2, and

tDNA-L4). GAPDH was used as a loading control at 36 kDa and the observed Mw of HPK1 protein was 90 kDa. **b)** Semi-quantitative analysis of the intensity of the bands shown in (a). Statistical significance was indicated as follows: *: $P < 0.05$, **: $P < 0.01$, ***: $P < 0.001$, ****: $P < 0.0001$, and n.s. indicates no significance. The error bars indicate the mean \pm SD values; $n = 3$. **c)** Fitted dose-degradation curve of the relationship between drug concentration and degradation in Jurkat cells. The DC_{50} value, representing the concentration at which 50% of the cell is degraded, was determined to be 262.0 nM. The error bars indicate the mean \pm SD values; $n = 3$.

After revision (lines 565-581): In addition, we explored the degradation of proteins located in different cellular compartments, including cytoplasmic hematopoietic progenitor kinase 1 (HPK1), which holds great promise for cancer immunotherapy. Thus, we developed novel HPK1-DbTACs that utilized a DNA tetrahedral scaffold co-conjugated with the azide-modified HPK1 ligand (L4) and CRBN ligand (L2). Through WB analysis (Supplementary Fig. 25a, b), a concentration-dependent degradation of HPK1 by these HPK1-DbTACs was observed, in contrast to the HPK1 kinase inhibitor sunitinib, which showed no degradation. Importantly, negative controls (Control, tDNA, tDNA-L2, and tDNA-L4) did not induce degradation either. Furthermore, the DC_{50} fit of HPK1 degradation was determined to be 262.0 nM (Supplementary Fig. 25c). These findings demonstrated the potential of DbTACs to target and degrade cytoplasmic proteins, expanding the application of the DbTACs platform.

After revision (lines 672-688): Fabrication of DbTACs and HPK1-DbTACs. The DNA strands were purchased from Sangon Biotech Co., Ltd. (Shanghai, China). The DNA sequences used were shown in Supplementary Table 1. To form DbTACs and HPK1-DbTACs, the single-stranded DNA S1 (50 μ M) was mixed with CDK9 ligand (500 μ M) or HPK1 ligand (500 μ M) at a 1:1 molar ratio. Additionally, DNA S4 (50 μ M) was mixed with CRBN ligand (500 μ M) at a 1:1 molar ratio. The two solutions were stirred at 28 $^{\circ}$ C for 2 h at 800 rpm/min, respectively. Subsequently, the mixtures were combined in equimolar ratios with pre-designed single-stranded DNA sequence S2 and S3 in TEM buffer (10 mM Tris, 1 mM EDTA, 20 mM $MgCl_2$, pH=8.0). The resulting solution was heated at 95 $^{\circ}$ C for 5 minutes and then annealed at 4 $^{\circ}$ C for at least 0.5 hours using a T series Multi-Block Thermal Cycler (LongGene, China) to stabilize the structures. Finally, the prepared DbTACs or bis-DbTACs were stored at 4 $^{\circ}$ C until further use.

After revision (lines 258-273):

General procedure for the synthesis of compounds HPK1 ligand (L4). The intermediate A-1 (0.117 g, 0.72 mmol), DIPEA (0.320 g, 2.48 mmol) and HATU (0.230 g, 0.60 mmol) were dissolved in 6 mL THF. The mixture was stirred at room temperature for 10 min. The compound 144 (0.241 g, 0.50 mmol) was added to the mixture and stirred at room temperature for 1 h. Then the reaction solution was extracted with DCM for three times. The combined extract was dried over anhydrous Na_2SO_4 . After filtration the solvent was evaporated under reduced pressure, and the residue was

purified on silica gel by flash column chromatography (CH₂Cl₂/MeOH, 80:1, vol/vol) giving compounds HPK1 ligand (200 mg, 64.3%) as white solid. ¹H NMR (300 MHz, CDCl₃) δ (ppm) = 11.13 (s, 1H), 8.59 (s, 1H), 8.36 (s, 1H), 7.69 (d, J = 6.0 Hz, 2H), 7.58 (t, J = 6.0 Hz, 3H), 7.51 (d, J = 9.0 Hz, 2H), 7.04 (d, J = 9.0 Hz, 2H), 3.82 (t, J = 4.5 Hz, 2H), 3.68-3.58 (m, 3H), 3.51-3.41 (m, 2H), 3.38-3.35 (m, 2H), 3.32-3.21 (m, 8H), 3.09 (s, 3H), 2.41 (t, J = 6.0 Hz, 2H), 1.98-1.87 (m, 2H), 1.77-1.81 (m, 4H), 1.51-1.43 (m, 2H). ¹³C NMR (75 MHz, DMSO-d₆) δ (ppm) = 171.30, 150.25, 148.16, 141.92, 136.25, 134.36, 131.23, 130.09, 128.26, 127.74, 126.84, 126.46, 123.63, 118.71, 116.98, 115.86, 58.70, 51.32, 49.63, 49.31, 45.45, 41.46, 33.04, 28.79, 26.61, 24.77. HRMS (ESI) m/z (M + H)⁺ calculated for C₃₅H₄₂N₈O₃: 623.3458; found, 623.3450.

3. On a similar note, to the question above, since the linker relies on nucleic acids, is there any risk of off-target effects? Are DNA binding proteins degraded when DbTACs are used? Could the authors comment on the chances of this happening? Similarly, could this be an advantage, and could the authors exploit their technology to target DNA-binding proteins more efficiently to degradation?

Response: Thank you for your professional and inspiring comments. Indeed, the use of nucleic acids as linkers in DbTACs introduces the potential risk of off-target effects, leading to the degradation of unintended DNA-binding proteins. According to this challenge, we would like to categorize DNA-binding proteins into two main categories based on their substrate sequences: those recognizing specific DNA sequences and those recognizing non-specific DNA sequences. For DNA-binding proteins that recognize specific DNA sequences, such as Toll 9, transcription factor NF-κB and GATA1, careful design of DbTACs to avoid these specific DNA sequences must be considered to minimize off-target effects. Additionally, for DNA-binding proteins that recognize non-specific DNA sequences, such as cyclic GMP-AMP synthase (cGAS) and Absent in Melanoma 2 (AIM2), steric hindrance, size, and morphology should also be taken into consideration. In this work, we carefully designed DNA tetrahedral sequences to avoid off-target binding. To address this concern, we performed a proteomics analysis of representative DbTACs-26 Å in the revised manuscript (Fig. 5a). The results demonstrated that non-obvious degradation of unintended DNA-binding protein was observed during the selective degradation of CDK9. This finding emphasizes the importance of sequence optimization to minimize off-target effects.

And yes this could also be an advantage. The modular nature of DbTACs allows for targeted degradation of DNA-binding proteins with increased efficiency. In this work, we utilized the intrinsic binding nature of DbTACs to efficiently target DNA-binding proteins, as exemplified by the successful design of Oligo-DbTACs against the ERG transcription factor (Fig. 8 in the Revised Manuscript). Through a modular approach, specific DNA motifs of ERG and ligand 2 of an E3 ligase were strategically incorporated into DNA tetrahedra to prepare Oligo-DbTACs, resulting in remarkable specificity for inducing ERG degradation, as evidenced by minimal effects observed with negative controls (NC and tDNA) (Fig. 8d). These findings highlight the potential of DbTACs technology in advancing our understanding and enabling more efficient degradation of target DNA-binding proteins.

In conclusion, DbTACs possess a dual nature, offering selective targeting of specific proteins while the risk of off-target effects. Careful consideration of DbTAC design, including substrate sequences, steric hindrance, size, and morphology, is crucial to mitigate off-target effects. Despite

this inherent risk, DbTACs offer a modular and customizable strategy for protein degradation, allowing for the optimization of design parameters to enhance selectivity and reduce off-target effects. Moreover, this modular approach provides a conceptual framework for the development of DNA nanostructure-based degraders with improved specificity and therapeutic potential. The above discussion has been added in the Revised Manuscript Page 10 lines 596-603.

Fig. 5a. Volcano plot showing fold changes of protein abundance from global proteomics analysis of MV4-11 cells treated with DbTACs-26 Å for 6 h at 200 nM.

Fig. 8. | Design, preparation, characterization, and efficacy of Oligo-DbTACs formed by DNA motif as POI ligand. **a**, Schematic illustration of the design strategy for Oligo-DbTACs formed using DNA tetrahedra as a linker conjugating L2 and ERG ligand, with binding moiety (purple) and three extra nucleotides (black) for protection of oligo degradation. **b**) Agarose gel electrophoresis and **c**) PAGE images of the self-assembly process of DbTACs formed using DNA motif as POI

ligand. **d**, WB analysis of the targeted degradation of ERG protein by DbTACs with different concentrations in PC3 cells. A non-specific sequence was chosen as the negative control (NC) target ligand. The observed Mw of ERG was 55 kDa, and that of GAPDH was 36 kDa. And the semiquantitative analysis of their grayscale. The error bars indicate the mean \pm SD values; n = 3. *: P<0.05, **: P<0.01.

4. Throughout the paper, one of the main controls missing is the use of the DNA tetrahedron alone, not labeled with any small molecule, protein or oligonucleotide. This control is key to demonstrate that the DNA architecture used is not responsible for some of the phenotypes and decrease in protein observed.

Response: Thank you for your valuable comments. We sincerely apologize for not including DNA tetrahedra alone as a control in our previous manuscript. Inspired by your comments, we have now incorporated DNA tetrahedra as a control in the western blot assays of DbTACs, bis-DbTACs, Abs-DbTACs, and Oligo-DbTACs. The updated results clearly demonstrate that the individual DNA tetrahedral structure (tDNA) does not contribute to the degradation of the target protein (Fig.3a, Fig.6d, Fig.7e, Fig.8d and Supplementary Fig.25a, b in revision). We have discussed these new findings in the Revised Manuscript Page 4 lines 226-229, Page 8 lines 454-457, Page 9 lines 525-527, Page 10 lines 558-560, and Page 10 lines 575-576 highlighting the pertinent information in yellow background. The above figures and corresponding discussion were attached as follows for your convenience. We sincerely appreciate your help in improving the comprehensiveness and scientific rigor of our study.

After revision (lines 226-229): Importantly, a DNA tetrahedral control (tDNA) demonstrated that the DNA architecture alone was not responsible for the observed protein degradation.

After revision (lines 454-457): Notably, the individual tDNA component was not responsible for the degradation of CDK6 or CDK9. Furthermore, neither of bis-DbTACs nor tDNA induced degradation of CDK1/2.

After revision (lines 525-527): Similarly, the individual tDNA exhibited negligible influence on the degradation process.

After revision (lines 558-560): The kinetics experiment confirmed that DbTACs, but not the NC or tDNA, effectively degraded ERG protein in a dose-dependent manner (Fig. 8d).

After revision (lines 575-576): Importantly, negative controls (Control, tDNA, tDNA-L2, and tDNA-L4) did not induce degradation either.

Response Fig. 2. The updated WB results of DbTACs, bis-DbTACs, Abs-DbTACs, Oligo-DbTACs, and HPK1-DbTACs in this study. The updated WB results included the DNA tetrahedron alone, not labeled with any small molecule (Fig. 3a, Fig. 6d, and Supplementary Fig. 25a, b), protein (Fig. 7e) or oligonucleotide (Fig. 8d).

5. Figure 2 and Lines 195-196: The use of gold nanoparticles clearly shows the localization of the two modifications on the DNA architecture. I wonder if tagging the DNA architecture with fluorophore and demonstrating the cell uptake could be beneficial and inform on the intracellular localization of DbTACs.

Response: We appreciate the reviewer's valuable suggestion regarding the intracellular localization of DbTACs. To address this concern, we performed additional experiments using a fluorescent labeling method to track the subcellular localization of DbTACs. Specifically, we labeled DbTACs-26 Å with Cy3 (DbTACs-26 Å-Cy3) and incubated them with HepG2 cells for various durations (0 min, 10 min, 3 h, 6 h, 12 h, and 24 h) (Supplementary Fig. 6). Nuclei were stained with DAPI to visualize the localization and inverted fluorescence microscopy was employed to observe the

fluorescence distribution.

The results revealed a robust Cy3 red fluorescence signal on the cell membrane after a 10-minute incubation, indicating the presence of DbTACs-26 Å-Cy3 at the cell surface. Over time, DbTACs-26 Å-Cy3 were progressively internalized by the cells and localized within the cytoplasm after 3 hours. Subsequently, DbTACs-26 Å-Cy3 accumulated around the nucleus. Notably, with extended incubation, DbTACs-26 Å-Cy3 eventually entered the nucleus. These findings shed light on the intracellular trafficking of DbTACs and have been included in the Revised Manuscript Page 4 lines 206-214 and the Revised Supplementary information Page 5 lines 43-48 with yellow background. A detailed method was added in the Revised Manuscript Page 12 lines 825-839.

We sincerely appreciate the valuable input provided by the reviewer, which has significantly enhanced the comprehensiveness and clarity of our study.

Supplementary Fig. 6. Subcellular localization of DbTACs-26 Å in HepG2 cells. HepG2 cells were treated with DbTACs-26 Å-Cy3 for different durations (0, 10 min, 3 h, 6 h, 12 h, and 24 h) and subsequently stained with DAPI. The red fluorescence represents the labeling of DbTACs-26 Å, while the blue color indicates the cell nucleus stained with DAPI. The Merge layer in the image displays the colocalization of DbTACs-26 Å, indicated by a white arrow. Scale bar: 20 µm.

6. Lines 200-210 and Figure 3.A): In the quantification of the Western Blot, all conditions have a significant error bar except for 21 and 26 Angstrom DbTACs. Is this discrepancy due to a difference in sample number? Have the experiments been replicated more times with these DbTACs than the other ones? The authors also claim better degradation than the PROTAC B11, while the WB shows similar levels of degradation between B11 and 26A DbTAC.

Response: Thank you for raising these important points regarding the quantification of Western Blot data and the comparison between DbTACs and PROTACs B11. We appreciate your attention to detail and have carefully considered your comments.

In terms of sample number, we confirm that the sample number was consistent across all experimental conditions. Each condition was replicated three times independently to ensure the

reliability of the results. According to your comment above, we have added individual DNA tetrahedral structure (tDNA) as a control group in this WB experiment, with each group replicated three times again. The results also showed that compared to other linker lengths, the degradation of DbTACs with 26 Å was best, which was comparable with PROTACs B11. And the error bar exhibited a difference. It is important to note that the Y-axis represents the relative gray value, normalized to the control group, ranging from 0 to 1. Although the error bars are visible, the actual numerical differences are relatively small. We apologize for any confusion caused by the presentation of the error bars in the figure, and we ensure that this is clarified with all bar graphs with plots that feature information about the distribution of the underlying data of updated Fig. 3a in the Revised Manuscript. And we provide additional clarification by including the description "The error bars indicate the mean \pm SD values; n = 3." in the Revised Manuscript.

Fig. 3a. The updated WB analysis and quantification of CDK9 protein levels in MV4-11 cells treated with 200 nM DbTACs with different linker lengths (DbTACs-8 Å, -11 Å, -16 Å, -21 Å, -26 Å, and -57 Å) and a positive compound B11 for 6 h. Observed Mw of CDK9 protein was 43 kDa, and GAPDH was used as a loading control at 36 kDa. ****: $P < 0.0001$, n.s. represents no significance. The error bars indicate the mean \pm SD values; n = 3.

Regarding the comparison of degradation effects between DbTACs-26 Å and PROTAC B11, the Western Blot images demonstrated similar degradation effects. We also performed a significant difference analysis, which showed no significant difference between the two groups. Consequently, we have changed the description as follows: "A human acute myeloid leukemia cancer cell line, MV4-11, was incubated with various DbTACs, and the degradation rates were analyzed using Western Blot (WB) (Fig. 3a). The results showed that DbTACs were excellent in inducing CDK9 degradation." in the Revised Manuscript Page 4 lines 216-221 with yellow background.

7. In some cases, e.g. Figure 3A and Fig 3C, it looks like some parts of the blot are darker or lighter. It would be great to add the full Western Blot image in the supporting information to make sure the decrease observed is not an artifact.

Response: Thank you for your suggestion, and we have included all complete Western Blot images in the original data file, available in Folder source-data. These images provide a more comprehensive view of the data and clearly demonstrate that the observed decrease in band intensity is not an artifact. These images are consistent with the quantification data presented in the main figures. We believe that the inclusion of these additional images will address the reviewer's concern and enhance the clarity of the manuscript.

8. Lines 272-276: The authors define cooperativity using KD^{ternary} twice instead of its real definition which is $= KD^{\text{ternary}} / KD^{\text{binary}}$.

Response: We sincerely appreciate the reviewer for bringing this oversight to our attention. We apologize for this error and any confusion it may have caused. We have revised the texts to correct KD^{ternary} as $KD^{\text{ternary}}/KD^{\text{binary}}$, specifically marked in the Revised Manuscript Page 5 line 304 and line 307 with yellow background. We sincerely appreciate the reviewer's valuable input, which has undoubtedly contributed to the improvement of our work.

9. Fig 4.d-g I am surprised by the low RU obtained in the SPR experiment. Based on the MW of the DbTACs, the theoretical maximum RU observed with 500 RU of CDK9 immobilized should be much higher. Is that small change due to a high reference binding from the DbTAC and the protein? On figure 4b, the change in retention time is also minimal despite the molecular weight almost doubling when both proteins are present.

Response: We appreciate the thoughtful comments and suggestions provided by the reviewer regarding Fig. 4d-g. The lower observed response unit (RU) values in the SPR experiment compared to the theoretical maximum RU based on the molecular weight (MW) of the DbTACs are indeed surprising. As you mentioned, non-specific binding or oxidation issues on the sensor surface could interfere with the molecule-surface interaction, leading to fluctuations in the reference binding. Although the reference binding should theoretically be 0, we observed fluctuations in the range of 4.8~7.5 RU, which might have affected the overall RU values (Response Fig. 3). We understand this as an important aspect to consider in our analysis, and we will carefully examine the sensor surface condition and optimize experimental procedures to minimize non-specific binding in future experiments.

Additionally, experimental conditions, molecular affinity, and molecular concentration can also contribute to the observed RU values. In our immobilization process, pH adjustments and protein preconcentration inadvertently led to some protein inactivation, which could have affected the RU values due to the sensitivity of CDK9 proteins to acidity. However, it is worth noting that despite these challenges, we successfully observed concentration-dependent trends and determined the affinity of the interaction, which remained unaffected. These findings align with the existing literature on low SPR responses, as demonstrated in Response Fig. 4 [*J Inflamm Res*, 2021, 14, 1455-1471; *Acta Pharm Sin B*, 2021, 11, 222-236; *PNAS*, 2017, 114, E5986-E5994; *ACS Cent Sci*, 2022, 8, 1102-1115]. We appreciate your comment and understand the need for more detailed explanation and optimization. Therefore, we will continue to follow up reports on preferable methods and techniques to address these limitations in our ongoing research. We apologize for the inadequate explanation of this in the manuscript.

Response Fig. 3. Reference bindings of samples. The reference bindings of binary complexes (DbTACs-26 Å) and ternary complexes (DbTACs-26 Å, DbTACs-8 Å, and DbTACs-57 Å preincubated with human recombinant CRBN protein).

Response Fig. 4. Representative SPR diagrams from relevant literature. These diagrams are obtained from the following sources: **a)** *J Inflamm Res*, 2021, 14, 1455-1471, **b)** *Acta Pharm Sin B*, 2021, 11, 222-236, **c)** *PNAS*, 2017, 114, E5986-E5994, and **d)** *ACS Cent Sci*, 2022, 8, 1102-1115.

We appreciate the reviewer's astute observation regarding the minimal change in retention time observed in Figure 4b, despite the almost doubling of the molecular weight when both proteins are present. In Figure 4b, we compared the retention time of DbTACs alone ($MW_{\text{DbTACs}} = 79.582 \text{ kDa}$) and DbTACs incubated with two proteins ($MW_{\text{DbTACs}} + MW_{\text{CDK9}} + MW_{\text{CRBN}} = 79.582 \text{ kDa} + 43 \text{ kDa} + 18.3 \text{ kDa} = 140.882 \text{ kDa}$). The difference in molecular weight is 61.3 kDa, resulting in a change in retention time of 1.808 min. Although the change in retention time is relatively small compared to the large molecular weight difference, it is important to consider that retention time is influenced by various factors, including molecular weight, mobile phase composition, pore size, and sample characteristics.

To further explore this surprising observation, we conducted tests using BSA (66 kDa) and antibody (150 kDa) samples with known molecular weights on the same chromatographic column (Guangzhou, China, Galaxy SEC S2000 5u, column size: 300*7.8mmI.D.). The SEC-HPLC results showed that a molecular weight difference of 84 kDa resulted in a retention time change of 0.72 min (Response Fig. 5). These comparisons indicate that large molecular weight differences may not show a significant difference of retention time. We recognize the need for improving the resolution and sensitivity of our chromatographic analysis. And we appreciate the reviewer's insightful comments and will carefully consider these factors in our future studies.

Response Fig. 5. Retention time analysis of samples. Retention times spectra of BSA (66 kDa), antibody (150 kDa), and a mixture of BSA and antibody.

10. Paragraph starting at line 370: Generalizability of DbTACs.

- a. The authors re-use CDK9 as a target using an antibody. It would have been interesting to target a protein without a small molecule ligand in this case.**
- b. In this paragraph, the authors do not comment on the linker length used to display the CRBN ligand. Is 26A still preferable? It is expected that the use of an antibody or oligonucleotide would change the shape of the ternary complex induced. Were different linker lengths tested to achieve a good level of degradation?**
- c. Does the presence of the antibody or the oligonucleotide affect the cell permeability of the DbTACs? Is the half-life of the antibody conjugate also 6h or is degradation by proteases or nucleases a concern?**

Response:

- a. The authors re-use CDK9 as a target using an antibody. It would have been interesting to target a protein without a small molecule ligand in this case.**

Thank you for providing valuable comments on our manuscript. We greatly appreciate your suggestion regarding the targeting of proteins without small-molecule ligands, as it would enhance the versatility of DbTACs. Indeed, in another research project, we utilize the self-assembly characteristic of DNA structure to achieve an intracellular in-situ synthesis of DbTACs. In this project, we target telomerase using an antibody as the ligand. Experimental results showed a

significant reduction in telomerase expression in U87MG cells, which overexpress telomerase (see Response Fig. 6). These findings ensure the potential of DbTACs platform for degrading targets within different ligands, and hope we can share this intracellular in-situ assemble DbTACs work with you soon.

Response Fig. 6. The degradation ability of the degraders in U87 cells. The changes in telomerase levels within U87 cells following treatment with the degraders (final concentration of 500 nM) were analyzed using western blot analysis.

b. In this paragraph, the authors do not comment on the linker length used to display the CRBN ligand. Is 26Å still preferable? It is expected that the use of an antibody or oligonucleotide would change the shape of the ternary complex induced. Were different linker lengths tested to achieve a good level of degradation?

Thank you for your insightful comments regarding the linker length used to display CRBN ligands in our study.

In our research, we investigated the degradation of CDK9 protein using small molecule ligands-based DbTACs with various linker lengths. Our results revealed that the 26 Å linker length demonstrated the highest binding affinity and degradation efficacy for small molecule ligands-based DbTACs. The small molecule ligands of CDK9 typically range in size from 0.3-0.5 nm. Meanwhile, the DNA tetrahedra framework used in our study had a size of 7.14 nm. Thus, studying the effect of ligand spacing in this case, particularly within frameworks like DNA tetrahedra, holds significant scientific relevance.

We apologize for not specifically addressing the different ligand lengths of DbTACs based on antibodies or oligonucleotides in this study. Nevertheless, it is important to note that when the size of the ligands, such as antibodies (10-15 nm) or oligonucleotides (6-8.5 nm), significantly exceeds that of the DNA tetrahedra framework (7.14 nm) utilized in our research, the precise ligand spacing may have a diminished impact. In this case, we recognize that the shape of the induced ternary complex can be mainly influenced when antibodies or oligonucleotides are employed as ligands. Therefore, investigating the effect of ligand orientation on protein degradation is a crucial aspect that we intend to address in our future studies. Fortunately, the double-helical nature of the DNA framework enables versatile localization of ligands in various directions, which will be a primary focus in our forthcoming research endeavors.

c. Does the presence of the antibody or the oligonucleotide affect the cell permeability of the DbTACs? Is the half-life of the antibody conjugate also 6h or is degradation by proteases or

nucleases a concern?

Thank you for your insightful comment. As mentioned in research [*Accounts of Chemical Research*, 2018, 51, 2305–2313], macromolecules such as antibodies and oligonucleotides generally have lower cell permeability compared to small molecules due to their large size. Therefore, to investigate the cellular uptake capacity of DbTACs-26 Å, Abs-DbTACs, and Oligo-DbTACs, fluorescence imaging experiments were performed. The compounds were labeled with Cy3 and incubated with HepG2 cells for 3 h and 12 h, respectively. As shown in Response Fig. 7, the cellular uptake of HepG2 cells was observed to be highest for DbTACs, followed by Oligo-DbTACs, and then Abs-DbTACs. Although when the ligand of DbTACs is an antibody or oligonucleotide, the cell uptake efficiency was weakened to some extent, it can still meet the requirements of sufficient cell uptake and efficient protein degradation.

Response Fig. 7. The cellular uptake capacity of DbTACs-26 Å, Abs-DbTACs, and Oligo-DbTACs. DbTACs-26 Å, Abs-DbTACs, and Oligo-DbTACs were labeled with Cy3 and incubated with HepG2 cells for 3 h and 12 h, respectively. Subsequently, stained with DAPI. The red fluorescence represents the labeling of DbTACs-26 Å, Abs-DbTACs, and Oligo-DbTACs, and the blue indicates the cell nucleus. The Merge layer in the image displays the colocalization of HepG2 cells with DbTACs-26 Å, Abs-DbTACs, and Oligo-DbTACs, respectively. Scale bar: 20 μm.

Thank you for raising the concern regarding the stability of Abs-DbTACs. We have addressed this issue by conducting stability experiments and incorporating the results into the revised manuscript. Abs-DbTACs were incubated with PBS, 10% FBS-contained 1640 medium, protease, or DNase I for different time points. Abs-DbTACs exhibited remarkable stability in PBS for up to 12 h (supplementary Fig. 22a) and in cell medium for up to 6 h (supplementary Fig. 22b), and showed resistance to protease (supplementary Fig. 22c) and DNase (supplementary Fig. 22d) degradation within 12 h.

These findings and corresponding discussion have been included in the Revised Manuscript Page 9 lines 503-507 and the Revised Supplementary information Page 13 lines 123-127 with yellow background. Detailed experimental methods have been included in the Revised Manuscript Page 12 lines 795-801. We sincerely appreciate the valuable input provided by the reviewer, which has significantly enhanced the comprehensiveness and clarity of our study.

Supplementary Fig. 22. Stability analysis of Abs-DbTACs. Stability analysis of Abs-DbTACs in (a) PBS, (b) 10%FBS (RPMI-1640), (c) Protease, and (d) DNase I at different time points (0, 1, 2, 4, 6, 12, and 24 h) by agarose gel electrophoresis. The red dotted lines indicate the same horizontal position of the gels.

11. In their Discussion on line 453-455, the authors claim that their technology is a “one for all” strategy. I am not sure the small diversity of proteins used here allow them to make that claim. The questions above remain and I would expect ternary complexes to be very different depending on which type of target protein binder they use.

Response: Thank you for your comment about our previous discussion on lines 453-455 (lines 453-455: The research presented here takes inspiration from a commonly used Chinese proverb, that is "all for one, and one for all", attempting to develop a universal platform for profiling of PROTACs). We appreciate your insight and acknowledge that the limited diversity of proteins tested in our study does not fully support the claim of a "one for all" strategy for the DbTACs platform.

In the revised manuscript, we have provided additional information regarding the application of the DbTACs strategy for the degradation of cytoplasmic HPK1 targets. We have explored different protein localizations (such as nucleus and cytoplasm), diverse target proteins with various functions (such as cell cycle-dependent kinases and protein kinases), and utilized different types of ligands (such as small molecules, antibodies, and oligonucleotides), demonstrating the versatility of our DbTACs strategy. However, it is important to note that the limited number of targets studied in our

research may impose certain limitations and potential overstatements. We are aware that there may be unresolved challenges and aspects that we have not comprehensively addressed.

Thus, the corresponding discussion has been revised as follows: "The research presented here takes inspiration from a commonly used Chinese proverb, "all for one, and one for all", attempting to develop a modular platform for PROTACs." This revision has been added in the Revised Manuscript Page 10 lines 583-585. We encourage the scientific community to further utilize our platform to expand its applications and identify any existing shortcomings.

We appreciate your valuable input, and we thank you for contributing to the clarity and accuracy of our manuscript.

12. Line 461: DbTACs are without a doubt an impressive and exciting new technology. The authors achieved degradation of various targets intracellularly. However, I do not think they can claim that DbTACs can serve as a tool for SAR in linker type or length for the design of heterobifunctional molecules. Indeed, in their model of ternary complexes, different faces of the protein interact together. The side of the protein facing each other in PROTACs are facing the DNA tetrahedron in DbTACs and the target protein interacts with the E3 ligase on one side.

Response: Thank you for your insightful comment regarding the claim made in the previous manuscript about DbTACs can serve as a tool for structure-activity relationship (SAR) analysis in linker type or length for heterobifunctional molecule design. We agree with you that the previous claim was overstated at present stage. As you pointed out, the side of the protein facing each other in PROTACs are facing the DNA tetrahedron in DbTACs and the target protein interacts with the E3 ligase on one side. To better understand your point, we have drawn a diagram (Response Fig. 8). In this study, we focused on the effective distance between the target protein and E3 ligase, which enables effective labeling of target protein with ubiquitination and following protein degradation. Our strategy provides robust distance control, which can be served as a powerful tool to investigate the impact of linker length on protein degradation. However, as your point, DbTACs may also influence the orientation of proteins facing each other, which maybe affect the efficiency of ubiquitination labeling. The most interesting thing is the double helix structure of DNA tetrahedra allows for orientation adjustment. We can easily adjust the orientation of target protein and E3 ligase by site-specific modification of ligands in adjacent nucleotides. We greatly appreciate your inspiring comment, and this interesting study aligns with our current research focus.

We have revised the sentence to better reflect the point of our study: "As an advantageous platform, DbTACs afforded resolution at the angstrom level for interrogating spatial distance-activity relationships." This revision can be found in the Revised Manuscript Page 10 lines 589-591 with yellow background.

Response Fig. 8. The orientation in the protein-protein interactions between a) PROTACs and b) DbTACs.

13. Line 473: It is not clear what the authors mean by the products require phase transfer. Please explain.

Response: Thank you for your comment. By "phase transfer", we are referring to the transfer of a chemical species from one phase (e.g., organic phase) to another phase (e.g., aqueous phase) using a transfer agent. In the context of our study, we want to clarify that traditional PROTAC molecules are typically insoluble in water and require the use of organic solvents during their synthesis. This solubility challenge makes it difficult to directly use traditional PROTACs in biological experiments.

However, the DbTACs developed in this study were primarily based on DNA sequences, which are water-soluble molecules. They can be easily dissolved in aqueous solutions or PBS without the need for an organic solvent. As a result, DbTACs do not require phase transfer during the experimental process. This feature simplifies the experimental procedures and reduces potential side effects associated with residual organic solvents.

To ensure clarity, we have revised the description as follows: "DbTACs platform addresses the poor water solubility. They were primarily based on DNA sequences, which are water-soluble molecules. They can be easily dissolved in aqueous solutions or PBS without the need for an organic solvent. As a result, DbTACs do not require solvent replacement during the experimental process. This feature simplifies the experimental procedures and reduces potential side effects associated with residual organic solvents." This revision can be found in Revised Manuscript Page 11 lines 614-621 with yellow background.

14. Lines 497-500: The authors claim that DbTACs can be synthesized in only two steps. However, this is only counting the steps necessary to form the DNA tetrahedron. The small molecule ligands, modified with reactive moieties still need to be synthesized. The gain of time is marginal compared to classical heterobifunctional molecules.

Response: Thank you for your insightful comment on lines 497-500 (lines 497-500: "DbTACs perfectly avoid complex synthetic routes of traditional PROTACs through simple bio-orthogonality and self-assembly, evaluating the reactions from >10 steps to only 2 steps"). We appreciate your clarification regarding the synthesis steps of DbTACs and the importance of considering the synthesis of small molecule ligands modified with reactive moieties. In our manuscript, we only

mentioned the formation of DNA tetrahedra and the biological orthogonality of DNA tetrahedra to ligands in the two-step synthesis description, which may have been misleading. We apologize for any confusion caused by our previous wording.

Upon careful consideration, we have revised the manuscript to provide a more accurate and comprehensive description of the synthesis steps involved in DbTACs. We now highlight the modular advantages of DbTACs over traditional PROTACs in terms of synthesis pathways. The corresponding discussion is as follows: "Compared to conventional PROTACs, DbTACs present a modular approach to generate degraders. This approach effectively reduces the number of steps involved in preparing the linker itself, as well as the subsequent coupling of the linker with the ligands." in the Revised Manuscript Page 11 lines 646-650 with yellow background.

15. Methods: In general, the method section is not detailed enough. In too many cases are conditions missing to reproduce these experiments.

For instance, Line 688 Fabrication of DbTACs: in this section, the concentrations used in the click reaction are not described, only the stoichiometry is reported. Or Line 944 SPR Binding Assay: The authors only reported running buffer without precising what this buffer is. Even if the buffer is commercial, it could be HEPES or PBS.

The authors need to improve the accuracy and the level of details of their methods to ensure that these results can be reproduced and the technology used in other labs.

Response: Thank you for your valuable comments on the Method section of our manuscript. We apologize for any lack of clarity or missing information in the original manuscript, and we have revised the whole Method section to provide more accurate and comprehensive details.

Specifically addressing your concerns:

1) Fabrication of DbTACs and bis-DbTACs: we have included the concentrations used in the click reaction, in addition to the stoichiometry, to facilitate reproducibility. Detailed methods were in the Revised Manuscript Page 11 lines 672-696 with yellow background.

After revision (lines 672-696): Fabrication of DbTACs and HPK1-DbTACs. The DNA strands were purchased from Sangon Biotech Co., Ltd. (Shanghai, China). The DNA sequences used were shown in Supplementary Table 1. To form DbTACs and HPK1-DbTACs, the single-stranded DNA S1 (50 μ M) was mixed with CDK9 ligand (500 μ M) or HPK1 ligand (500 μ M) at a 1:1 molar ratio. Additionally, DNA S4 (50 μ M) was mixed with CRBN ligand (500 μ M) at a 1:1 molar ratio. The two solutions were stirred at 28 $^{\circ}$ C for 2 h at 800 rpm/min, respectively. Subsequently, the mixtures were combined in equimolar ratios with pre-designed single-stranded DNA sequence S2 and S3 in TEM buffer (10 mM Tris, 1 mM EDTA, 20 mM MgCl₂, pH=8.0). The resulting solution was heated at 95 $^{\circ}$ C for 5 minutes and then annealed at 4 $^{\circ}$ C for at least 0.5 hours using a T series Multi-Block Thermal Cycler (LongGene, China) to stabilize the structures. Finally, the prepared DbTACs or bis-DbTACs were stored at 4 $^{\circ}$ C until further use.

Fabrication of bis-DbTACs. The DNA sequences used were shown in Supplementary Table 2. For the formation of bis-DbTACs, the single-stranded DNA S1 (50 μ M), S2 (50 μ M), and S4 (50 μ M) were mixed with CDK9 ligand (500 μ M), CDK6 ligand (500 μ M), and CRBN ligand (500 μ M) at a 1:1 molar ratio, respectively. The three solutions were stirred at 28 $^{\circ}$ C for 2 h at 800 rpm/min. The subsequent steps are described as mentioned above.

2) SPR Binding Assay: we now specify the composition of the running buffer as PBS-P (contain 0.5% surfactant P20), providing the necessary information for replication. A detailed method was in the Revised Manuscript on Page 15 lines 1193-1214 with yellow background.

After revision (lines 1193-1214): SPR binding assay. For binary binding experiments, stock solutions of DbTACs-26 Å or free CRBN protein were serially diluted in PBS-P (containing 0.5% Surfactant P20) running buffer (twofold serial dilution). The diluted solutions were injected over a CM5 chip coated with immobilized CDK9 (~500 RU). For ternary binding study, DbTACs-8 Å/DbTACs-26 Å/DbTACs-57 Å were mixed with a solution of CRBN protein to prepare a final solution of 400 nM DbTACs and 800 nM CRBN protein in PBS-P running buffer. The complexes were preincubated in PBS-P running buffer for 0.5 h, followed by serial dilutions (six-point twofold serial dilutions).

For ternary binding study, DbTACs-8 Å/DbTACs-26 Å/DbTACs-57 Å were mixed with a solution of CRBN protein to prepare a final solution of 400 nM DbTACs and 800 nM CRBN protein in PBS-P running buffer. The complexes were preincubated in PBS-P running buffer for 0.5 h, followed by serial dilutions (six-point twofold serial dilutions).

SPR binding responses for binary and ternary complexes were performed in multicycle kinetic mode at 298.15 K with a contact time of 60 s, a flow rate of 30 $\mu\text{L min}^{-1}$ and a dissociation time of 60 s. The raw sensorgram data was processed using Biacore T200 Evaluation Software. The reference surface and blank injections were subtracted from the raw data before data analysis. Steady state affinity (SSA) model was used to calculate the association rate (k_{on}), dissociation rate (k_{off}) and dissociation constant (KD) for the binding affinity between binary and ternary complexes.

3) In addition, other details of the whole Method section were revised. For example, we have revised All-atom models of DbTACs (Revised Manuscript Page 15 lines 1119-1151) and included Molecular docking analysis (Revised Manuscript Page 15 lines 1152-1192 and Page 17 lines 1382-1389) with yellow background.

After revision (lines 1119-1151): All-atom models of DbTACs. All-atom models of various DbTACs, bis-DbTACs, Abs-DbTACs, Oligo-DbTACs, etc, were constructed using the PolygenDNA program^{43,44} and MOE software. To generate an all-atom model of a DNA tetrahedra using the PolygenDNA program, the DNA sequence of interest was specified as input. A double-stranded DNA helix was then generated, and the tetrahedral vertices were placed at the desired positions in 3D space. A series of energy minimization and geometry optimization steps were applied to refine the initial atom positions and adjust the geometry of the helix to achieve optimal bond lengths, bond angles, and dihedral angles. The resulting pdb file contained all-atom coordinates for all atoms in the DNA tetrahedron, including hydrogen atoms and other small molecules. Molecular visualization software, such as MOE, was used to visualize and analyze the model and make any necessary adjustments. To generate the final pdb file of series DbTACs by covalently linking the DNA tetrahedron and ligands, MOE was used. First, the 3D coordinates of the ligands were generated and optimized using the MOE Builder module. Next, the DNA tetrahedra pdb file was loaded into MOE, and the 3D coordinates of the DNA tetrahedra were optimized using the MOE Protein Preparation Wizard. The ligands were then docked into the optimized DNA tetrahedra structure using the MOE Dock module, and the covalent bonds between the ligands and DNA tetrahedra were formed using the MOE Editor module. Finally, the resulting structure was energy minimized using the MOE Energy Minimization module to obtain the all-atom model of

various DbTACs. MOE employs various algorithms to minimize the energy of the structure and optimize the geometry of the molecule. The optimized pdb file of various DbTACs was used for subsequent analysis and simulations.

After revision (lines 1152-1192): Molecular docking analysis. The molecular docking studies of DbTACs with varying linker lengths-CDK9 protein and DbTACs with varying linker lengths-CDK9-CRBN proteins were performed using the MOE software package, developed by Chemical Computing Group. The crystal structures of CDK9 protein (pdb ID: 3BLH) and CRBN protein (pdb ID: 4CI3) were downloaded from the Protein Data Bank (PDB) and prepared using the MOE Protein Preparation Wizard.

For the DbTACs with varying linker lengths-CDK9 docking, the DbTACs pdb file generated from the previous step was loaded into MOE. Prior to docking, the protein underwent preprocessing using QuickPrep with default options, which included rectifying structural inaccuracies, incorporating hydrogen atoms, optimizing three-dimensional H-bonding networks, eliminating water molecules beyond 4.5 Å from the protein, and minimizing within a limited range of 8 Å of the altered base pairs. Next, the CDK9 ligand on the DbTACs was identified as the Ligand Site using the MOE Site Finder module, and a docking box was defined around the site. The native ligand pockets of CDK9 protein were selected as the Receptor Site in the docking studies. The MOE Dock module was then used to dock DbTACs into the binding site of CDK9 using a Refinement with Rigid Body protocol, which generated several docking poses based on the predicted interaction energies between the ligand and receptor. The docking poses were ranked according to their binding affinities, and their Docking Score S^{45} was recorded. The pose with the highest Docking Score was selected as the final result. The Docking score S was calculated based on the following formula (1):

$$\Delta G_{Binding}^{Calc.} = \alpha \left(\frac{\frac{2}{3}(E_{Inter}^{Coul.} + \Delta G_{Bind}^R)}{\Delta G_{Bind}^{Elec.}} + \frac{E_{Inter}^{vdW} + \Delta G_{Bind}^{npsol}}{\Delta G_{Bind}^{Non-pola}} \right) + c \quad (1)$$

The Docking scoring S was primarily composed of $\Delta G_{Binding}^{Calc.}$, which included $\Delta G_{Bind}^{Elec.}$ and $\Delta G_{Bind}^{Non-pola}$. $E_{Inter}^{Coul.}$ and E_{Inter}^{vdW} represented the columbic and van der Waals contribution to binding, respectively. ΔG_{Bind}^R was the change in reaction field energy upon binding. The ΔG_{Bind}^{npsol} term represented the change in non-polar solvation (van der Waals and cavitation cost) upon binding. Furthermore, the scaling factor for electrostatic interactions was empirically determined to be 2/3, which improved accuracy compared to the theoretically ideal value of 1/2⁴⁶.

After revision (lines 1382-1389):

45. Corbeil, C.R., Williams, C.I. & Labute, P. Variability in docking success rates due to dataset preparation. *Journal of Computer-Aided Molecular Design* **26**, 775-786 (2012).
46. Ben-Amotz, D. & Underwood, R. Unraveling water's entropic mysteries: a unified view of nonpolar, polar, and ionic hydration. *Accounts of chemical research* **41**, 957-967 (2008).

16. Supplemental Information, Small molecule analysis: The analytical data for the small molecules synthesized is minimal. The authors provide 1H NMR and HRMS, for a journal like Nature Communications, it would be good to provide 13C for each small molecule synthesized. In the HRMS spectrum, the X-axis is really zoomed in for two out of the three compounds reported. The authors should provide the full spectrum and explain what the

smaller intensities peaks observed are. Overall, this level of characterization is insufficient. Additional HPLC traces could also improve the quality of the analysis.

Response: Thank you for bringing this issue to our attention. We apologize for the lack of sufficient analytical data for the small molecules synthesized in the Supplemental Information. In order to address this concern, we have made the following revisions:

We have now included ^1H NMR, ^{13}C NMR, HRMS, and HPLC data of each synthesized small molecule. This expanded characterization will provide a more comprehensive analysis of the chemical structure and purity of the compounds.

The HPLC analysis demonstrates that the purity of all compounds is above 95%, further ensuring the reliability and quality of the synthesized small molecules.

In the HRMS spectra, we have included the full spectrum and explained the presence of smaller intensity peaks, such as sodium peaks or double hydrogenation peaks. This additional information will help readers interpret the spectra accurately.

Please find our detailed spectra in the Revised Supplementary information Pages 25-40. These revisions address the insufficiency of the previous characterization and enhance the overall quality and reliability of our analysis. We sincerely appreciate your insightful comments, as they have contributed significantly to improving our manuscript. Thank you for bringing these concerns to our attention.

Reviewer 2

Comments:

In general, the potential of PROTACs to modify targets (POIs) thought to be unreachable, such as because they lack well-defined binding sites, and to down-regulate all of the POI's activities, has sparked a great deal of interest in this approach. E3 ligase ligands for either von Hippel-Lindau (VHL) or Cereblon (CRBN) are present in the majority of PROTACs identified in the literature. All PROTACs, with the exception of one, that have started clinical trials are based on CRBN, and this makes CRBN-based PROTACs very interesting.

In this paper, Li Zhou et al have developed a DbTACs-based approach, where smart DNA frameworks are used to combine substrates, providing a single universal platform for tackling many challenges. This intriguing technique not only sheds light on generic degrader design principles, but it also offers up new avenues for drug development in precision medicine. Briefly, DNA tetrahedra were used as templates in this study, and the spatial locations of each atom were specified. Consequently, by accurately positioning POI and E3 ligase ligands on templates using the computer models, ligand spacings are controlled from 8 to 57 Å and the ideal linker length (26 Å) between ligands achieve a high degradation rate (70%) and higher binding affinity ($K_{DA} = 42$ nM). Moreover, bispecific DbTACs (bis-DbTACs) accomplished multi-target depletion while preserving highly selective degradation of protein subtypes by combining trivalent ligands in a single DbTACs platform. This proof-of-concept study on DbTACs also applied to different types of warheads (for example, small molecules, antibodies, or DNA motifs). This reviewer is a PROTAC computational design expert, so my comments will mostly address the simulation part. However, I must stress that, in spite of the drawbacks mentioned below, all sections are clearly written and understandable for the broad readership of Nature Communication. The subject of the paper is quite important, and the approach for the computational design of DbTACs is very interesting. However, in my opinion, the paper suffers from some critical issues (as detailed below) that prevent its publication at the current stage. Owing to the importance of the theme and the suitability of the approach, I believe the authors should be given a chance to perform a major revision on the manuscript addressing those critical points.

The details provided for the different POI and E3L complexes from the modeling study in the manuscript and supporting information are not clear, and it is hard to follow and reproduce the work reported in the manuscript, at least using the current description. Therefore, authors should add more description on how the POI and E3L were docked and further extend the validation procedure on ranking of docking poses using the different linker lengths. In several places in the manuscript, the author refers to computational models, including in lines 172-175 of Figure 8, but it was hard to find enough detail in the method section and in the supporting information. In order to reproduce or to re-create this model for different POI with appropriate linker length, computational detail is important.

Response: Thank you for your valuable comments. We appreciate your expertise in PROTACs computational design and agree that additional details on the docking procedure for the POI and E3L complexes are necessary for reproducibility. We have made the following revisions to address these concerns:

We have provided a more detailed description of how the POI and E3L complexes were docked,

and two references were added to support the detailed molecular docking process. Specifically, we have outlined the methodology used to generate all-atom models of various DbTACs using the PolygenDNA program and MOE software. This includes the steps of constructing the DNA tetrahedra model, refining the atom positions, and covalently linking the DNA tetrahedra with the ligands. The revised description can be found in the Revised Manuscript Page 15 lines 1119-1151 with yellow background. Further, we have also provided a more detailed description of the molecular docking process of DbTACs with proteins. Molecular docking studies were performed using the MOE software package to investigate the interaction between DbTACs with varying linker lengths and CDK9 protein. The docking protocol involved protein preprocessing, site identification, docking box definition, and pose ranking based on binding affinities using the Dock module. The Docking Score (S) was calculated using a formula that incorporated various energy contributions, including electrostatic and non-polar interactions. We also added two references to support the reliability of molecular docking. A detailed method was revised in the Revised Manuscript Page 15 lines 1152-1192 and Page 17 lines 1382-1389 with yellow background. This section outlines the steps taken in the docking simulations, ensuring transparency and reproducibility of the computational models.

To further validate the docking poses and the impact of linker lengths, we have performed additional docking simulations using different DbTACs with various linker lengths (DbTACs-8 Å, -11 Å, -16 Å, -21 Å, and -57 Å). The docking scores were compared, and the results were included in the Revised Supplementary information Supplementary Fig. 10. Among the various linker lengths investigated, DbTACs-26 Å displayed the most stable ternary conformation, with a docking score of -80.59 (Fig. 4c in Revised Manuscript). In contrast, the other linker lengths exhibited relatively lower docking scores: DbTACs-8 Å: -73.39, DbTACs-11 Å: -64.90, DbTACs-16 Å: -64.01, DbTACs-21 Å: -63.54, and DbTACs-57 Å: 75.63 (Supplementary Fig. 10). This observation suggested that DbTACs-26 Å tended to form more stable ternary complexes. We discussed the implications of these docking results, highlighting the preferred linker length for optimal binding and stabilization of the complex. The corresponding discussion was added in the Revised Manuscript Page 5 lines 287-295.

We believe these revisions address the critical points you raised and provide the necessary computational details for the reproducibility and re-creation of the models. We appreciate your thorough review and valuable comments, as they have significantly contributed to improving the clarity and rigor of our manuscript. Thank you for giving us the opportunity to revise our work.

Supplementary Fig. 10. Molecular docking of DbTACs. The molecular docking images of (i) binary and (ii) ternary complex. Gray represents DbTACs, docked at the binding sites of the target protein. Yellow represents the CEBN E3 ligase and purple represents the CDK9 protein.

Please note that the detailed method of molecular docking of DbTACs have been marked in yellow in the revised manuscript as follows:

After revision (lines 1119-1151): All-atom models of DbTACs. All-atom models of various DbTACs, bis-DbTACs, Abs-DbTACs, Oligo-DbTACs, etc, were constructed using the PolygenDNA program^{43,44} and MOE software. To generate an all-atom model of a DNA tetrahedra using the PolygenDNA program, the DNA sequence of interest was specified as input. A double-stranded DNA helix was then generated, and the tetrahedral vertices were placed at the desired positions in 3D space. A series of energy minimization and geometry optimization steps were applied to refine the initial atom positions and adjust the geometry of the helix to achieve optimal bond lengths, bond angles, and dihedral angles. The resulting pdb file contained all-atom coordinates for all atoms in the DNA tetrahedron, including hydrogen atoms and other small molecules. Molecular visualization software, such as MOE, was used to visualize and analyze the model and make any necessary adjustments. To generate the final pdb file of series DbTACs by covalently linking the DNA tetrahedron and ligands, MOE was used. First, the 3D coordinates of the ligands were generated and optimized using the MOE Builder module. Next, the DNA tetrahedra pdb file was loaded into MOE, and the 3D coordinates of the DNA tetrahedra were optimized using the MOE Protein Preparation Wizard. The ligands were then docked into the optimized DNA tetrahedra structure using the MOE Dock module, and the covalent bonds between the ligands and DNA tetrahedra were formed using the MOE Editor module. Finally, the resulting structure was energy minimized using the MOE Energy Minimization module to obtain the all-atom model of various DbTACs. MOE employs various algorithms to minimize the energy of the structure and optimize the geometry of the molecule. The optimized pdb file of various DbTACs was used for subsequent analysis and simulations.

After revision (lines 1152-1192): Molecular docking analysis. The molecular docking studies of DbTACs with varying linker lengths-CDK9 protein and DbTACs with varying linker lengths-CDK9-CRBN proteins were performed using the MOE software package, developed by Chemical Computing Group. The crystal structures of CDK9 protein (pdb ID: 3BLH) and CRBN protein (pdb ID: 4CI3) were downloaded from the Protein Data Bank (PDB) and prepared using the MOE Protein Preparation Wizard.

For the DbTACs with varying linker lengths-CDK9 docking, the DbTACs pdb file generated from the previous step was loaded into MOE. Prior to docking, the protein underwent preprocessing using QuickPrep with default options, which included rectifying structural inaccuracies, incorporating hydrogen atoms, optimizing three-dimensional H-bonding networks, eliminating water molecules beyond 4.5 Å from the protein, and minimizing within a limited range of 8 Å of the altered base pairs. Next, the CDK9 ligand on the DbTACs was identified as the Ligand Site using the MOE Site Finder module, and a docking box was defined around the site. The native ligand pockets of CDK9 protein were selected as the Receptor Site in the docking studies. The MOE Dock module was then used to dock DbTACs into the binding site of CDK9 using a Refinement with Rigid Body protocol, which generated several docking poses based on the predicted interaction energies between the ligand and receptor. The docking poses were ranked according to their binding affinities, and their

Docking Score S^{45} was recorded. The pose with the highest Docking Score was selected as the final result. The Docking score S was calculated based on the following formula (1):

$$\Delta G_{Binding}^{Calc.} = \alpha \left(\frac{2}{3} \frac{(E_{Inter}^{Coul.} + \Delta G_{Bind}^R)}{\Delta G_{Bind}^{Elec.}} + \frac{E_{Inter}^{vdW} + \Delta G_{Bind}^{npsol}}{\Delta G_{Bind}^{Non-polar}} \right) + c \quad (1)$$

The Docking scoring S was primarily composed of $\Delta G_{Binding}^{Calc.}$, which included $\Delta G_{Bind}^{Elec.}$ and $\Delta G_{Bind}^{Non-polar}$. $E_{Inter}^{Coul.}$ and E_{Inter}^{vdW} represented the columbic and van der Waals contribution to binding, respectively. ΔG_{Bind}^R was the change in reaction field energy upon binding. The ΔG_{Bind}^{npsol} term represented the change in non-polar solvation (van der Waals and cavitation cost) upon binding. Furthermore, the scaling factor for electrostatic interactions was empirically determined to be $2/3$, which improved accuracy compared to the theoretically ideal value of $1/2^{46}$.

After revision (lines 1382-1389):

45. Corbeil, C.R., Williams, C.I. & Labute, P. Variability in docking success rates due to dataset preparation. *Journal of Computer-Aided Molecular Design* **26**, 775-786 (2012).
46. Ben - Amotz, D. & Underwood, R. Unraveling water's entropic mysteries: a unified view of nonpolar, polar, and ionic hydration. *Accounts of chemical research* **41**, 957-967 (2008).

Reviewer: 3**Comments:**

In this study, the authors have developed DNA framework-engineered chimeras platform to enable selectively targeted protein degradation, for which they named DbTACs. DNA tetrahedra were employed as templates for the synthesis of PROTAC degraders by precisely locating ligands of POI and E3 ligase on templates, ligand spacings were controllably manipulated from 8 Å to 57 Å. In one example, the authors showed that their designed DbTAC degrader is effective in reducing CDK9 protein by 70% when the linker length is 26 Å. Impressively, this degrader also shows selectivity over other CDK proteins. The authors further showed that multitargeted DbTAC degraders can be designed, using CDK6 and CDK9 as examples. The authors further demonstrated that ligands for POIs can be extended to antibody or DNAs using an antibody against CDK9 and DNAs targeting ERG.

Overall, this is a very interesting study, suggesting that the DNA framework-engineered chimeras platform can be used as a general platform to target many proteins using different types of ligands. Hence, it is recommended that the paper be published in Nature Communications with the following revisions.

First, in the TPD field, proteomics has been widely used to demonstrate the selectivity of designed degraders. It is recommended that the authors perform a proteomics analysis at least on one representative degrader to demonstrate its selectivity more broadly.

Second, protein degradation certainly leads to phenotypes in cells. It is recommended that the authors show the phenotypes of protein degradation with at least some of those degraders shown in this study.

Response:

1. First, in the TPD field, proteomics has been widely used to demonstrate the selectivity of designed degraders. It is recommended that the authors perform a proteomics analysis at least on one representative degrader to demonstrate its selectivity more broadly.

Thank you for your valuable comment. We appreciate your suggestion regarding the use of proteomics analysis to further demonstrate the selectivity of our degraders. In response to this concern, we have conducted a proteomics analysis on one representative degrader, DbTACs-26 Å, with a specific focus on its target CDK9. The analysis was performed on MV4-11 cells from different treatment groups using 4D-FastDIA quantitative proteomic analysis, providing a comprehensive view of the selective degradation mechanism induced by DbTACs-26 Å. To ensure high-quality analysis results, the dataset obtained from this analysis comprised 43,504 peptides, 6,294 identified proteins, and 6,275 quantifiable proteins (Supplementary Fig. 16).

The proteomics analysis revealed distinct protein expression patterns between the control group and the DbTACs-26 Å group, as demonstrated by principal component analysis (PCA) (Supplementary Fig. 17). Among the differentially expressed proteins, we observed 11 upregulated proteins and 131 downregulated proteins in the DbTACs-26 Å group compared to the control group (Fig. 5a). Notably, among the down-regulated proteins, CDK9 protein exhibited the most significant downregulation in abundance, while CDK1/2 and CDK6 proteins were not down-regulated. These results are consistent with the aforementioned cellular experiments. To investigate the selective degradation mechanism of DbTACs, subcellular distribution analysis (Fig. 5b) and cellular component analysis (Fig. 5c) of these differentially expressed proteins was performed. The results

indicated a prominent localization of these differentially expressed proteins in the nucleus, supporting the targeting of nuclear proteins by DbTACs-26 Å. Further functional analysis, including gene ontology (GO) enrichment analysis, highlighted the involvement of differentially expressed proteins in critical biological processes related to DNA damage and cell cycle processes (Fig. 5d). Importantly, the molecular function analysis emphasized the impact on cyclin-dependent protein kinase activity, consistent with the targeted degradation of CDK9 by DbTACs-26 Å (Fig. 5e). Furthermore, the KEGG pathway analysis revealed the p53 signaling pathway as a key mechanism associated with the significantly changed proteins in the DbTACs-26 Å group (Fig. 5f). Protein-protein interaction analysis further demonstrated interactions among different proteins, including CDK9, and their association with TP53, supporting the selective degradation mechanism of DbTACs-26 Å (Fig. 5g).

The results of the proteomics analysis are consistent with our previous cellular assays, further supporting the selectivity of DbTACs-26 Å towards its intended target, CDK9. These findings also provide valuable insights into the selective degradation mechanism of DbTACs. The proteomics analysis data, along with a comprehensive discussion, has been incorporated into the Revised Manuscript Page 7 lines 385-422 and in the Revised Supplementary information Page 10-11 lines 97-104 with yellow background. Detailed information about the proteomics analysis method has also been provided in the Revised Manuscript Page 14 lines 1016-1098 with yellow background. The above data, discussion and method were attached as follows for your convenience.

We sincerely appreciate your suggestion and believe that the inclusion of this data strengthens the overall impact of our work.

Fig. 5. | Proteomics analysis of differential protein in MV4-11 cells treated by DbTACs-26 Å. a, Volcano plot showing fold changes of protein abundance from global proteomics analysis of MV4-11 cells treated with DbTACs-26 Å for 6 h at 200 nM. **b,** Subcellular localization prediction of identified proteins using WoLFPSORT. The subcellular localization of identified proteins was predicated using

WoLFPSORT database with amino acid sequences of identified proteins. **c**, Gene Ontology (GO) analysis of cellular component of samples between DbTACs-26 Å-treated group and control group. Molecular function analysis of between compound DbTACs-26 Å-treated group and control group. Cluster analysis of **d**) biological process, **e**) molecular function, and **f**) KEGG pathway of samples between DbTACs-26 Å-treated group and control group. n=3. **g**, Protein-protein interaction network analysis of CDK9 with other proteins.

Supplementary Fig. 16. Quality check of proteomic data. a) The overview of all identified proteins. b) The distribution of peptides length among the identified peptides after trypsin digestion. c) The distribution of the number of peptides contained in the proteins.

Supplementary Fig. 17. Principal component analysis (PCA) analysis. PCA analysis in MV4-11 cells treated with DbTACs-26 Å or control groups. The analysis was conducted using a dataset comprising replicates from three independent experiments (n=3).

Please note that the changes have been marked in yellow in the revision:

After revision (lines 385-422): To further investigate the selective degradation mechanism of DbTACs, 4D-FastDIA quantitative proteomic analysis was performed on the MV4-11 cells from different treatment groups. Based on these qualified data (Supplementary Fig. 16), a total of 43,504 peptides, 6,294 proteins, and 6,275 quantifiable proteins were identified in MV4-11 cells. Principal component analysis (PCA) revealed distinct protein expression patterns between the control group and the DbTACs-26 Å group, and they were relatively separated from each other (Supplementary Fig. 17). Among the proteins exhibiting significant changes, the DbTACs-26 Å group displayed 11

upregulated proteins and 131 downregulated proteins (Fig. 5a). Notably, among the down-regulated proteins, CDK9 protein exhibited the most significant downregulation in abundance, while CDK1/2 and CDK6 proteins were not down-regulated. These results are consistent with the aforementioned cellular experiments. To investigate the potential functional enrichment of the differentially expressed proteins, subcellular distribution analysis (Fig. 5b) and cellular component analysis (Fig. 5c) were performed on these proteins. The results indicated prominent localization of these differentially expressed proteins in the nucleus, supporting the targeting of nuclear proteins by DbTACs-26 Å. Further functional analysis, including gene ontology (GO) enrichment analysis, highlighted the involvement of differentially expressed proteins in critical biological processes related to DNA damage and cell cycle processes (Fig. 5d). Importantly, the molecular function analysis emphasized the impact on cyclin-dependent protein kinase activity, aligning with targeted degradation of CDK9 by DbTACs-26 Å (Fig. 5e). Furthermore, the KEGG pathway analysis revealed the p53 signaling pathway as a key mechanism associated with the significantly changed proteins in the DbTACs-26 Å group (Fig. 5f). Protein-protein interaction analysis further demonstrated interactions among different proteins, including CDK9, and their association with TP53, supporting the selective degradation mechanism of DbTACs-26 Å (Fig. 5g).

After revision (lines 1016-1098): Quantitative proteomics analysis. The 4D-FastDIA-based quantitative proteomic analysis of human MV4-11 cells was carried out by Jingjie PTM Biolabs Inc. (Hangzhou, China). Samples were sonicated three times on ice using a high intensity ultrasonic processor (Scientz) in lysis buffer (8 M urea, 1% protease inhibitor cocktail). The remaining debris was removed by centrifugation at 12,000 g at 4 °C for 10 min. Finally, the supernatant was collected and the protein concentration was determined with BCA kit according to the manufacturer's instructions.

For digestion, the protein solution was reduced with 5 mM dithiothreitol for 30 min at 56 °C and alkylated with 11 mM iodoacetamide for 15 min at room temperature in darkness. The protein sample was then diluted by adding 100 mM TEAB to urea concentration less than 2 M. Finally, trypsin was added at 1:50 trypsin-to-protein mass ratio for the first digestion overnight and 1:100 trypsin-to-protein mass ratio for a second 4 h-digestion. Finally, the peptides were desalted by C18 SPE column.

The tryptic peptides were dissolved in solvent A (0.1% formic acid, 2% acetonitrile/in water), directly loaded onto a home-made reversed-phase analytical column (25-cm length, 75/100 µm i.d.). Peptides were separated with a gradient from 6% to 24% solvent B (0.1% formic acid in acetonitrile) over 70 min, 24% to 35% in 14 min and climbing to 80% in 3 min then holding at 80% for the last 3 min, all at a constant flow rate of 450 nL/min on a nanoElute UHPLC system (Bruker Daltonics).

The peptides were subjected to capillary source followed by the timsTOF Pro (Bruker Daltonics) mass spectrometry. The electrospray voltage applied was 1.60 kV. Precursors and fragments were analyzed at the TOF detector, with a MS/MS scan range from 100 to 1700 m/z. The timsTOF Pro was operated in parallel accumulation serial fragmentation (PASEF) mode. Precursors with charge states 0 to 5 were selected for fragmentation, and 10 PASEF-MS/MS scans were acquired per cycle. The dynamic exclusion was set to 30 s.

The resulting MS/MS data were processed using MaxQuant search engine (v.1.6.15.0). Tandem mass spectra were searched against the human SwissProt database (20422 entries) concatenated with reverse decoy database. Trypsin/P was specified as cleavage enzyme allowing up to 2 missing cleavages. The mass tolerance for precursor ions was set as 20 ppm in first search and 5 ppm in

main search, and the mass tolerance for fragment ions was set as 0.02 Da. Carbamidomethyl on Cys was specified as fixed modification, and acetylation on protein N-terminal and oxidation on Met were specified as variable modifications. FDR was adjusted to < 1%.

Then, principal component analysis (PCA) was used to evaluate the repeatability of samples from each group. Differential proteins were obtained after the qualification of samples, whose differences in relative quantification in two groups were compared by T-test, and the corresponding p-value was calculated. In addition, with a criterion of p-value ≤ 0.05 , the protein ratio > 1.5 was regarded as up-regulation, while the protein ratio $< 1/1.5$ as down-regulation.

Based on the identified proteins, the subcellular localization analysis was performed using WoLF-PSORT database; GO annotation is to annotate and analyze the identified proteins with eggno-mapper software (v2.1.6). The software is based on the EggNOG database (v5.0.2, <http://eggno5.embl.de/#/app/home>). Extracting the GO ID from the results of each protein note, and then classified the protein according to Cellular Component, Molecular Function, and Biological Process; Kyoto Encyclopedia of Genes and Genomes (KEGG) database (v5.0, <http://www.kegg.jp/kegg/mapper.html>) was used for KEGG pathway enrichment analysis. Fisher's exact test was used to analyze the significance of KEGG pathway enrichment of differentially expressed proteins (using the identified protein as the background), and P value < 0.05 were considered significant. Furthermore, all differentially expressed protein database accession or sequence were searched against the STRING database (v11.5, <https://cn.string-db.org/>) for protein-protein interactions. Only interactions between the proteins belonging to the searched data set were selected, thereby excluding external candidates. STRING defines a metric called "confidence score" to define protein-protein interaction (PPI) confidence; we fetched all interactions that had a confidence score ≥ 0.7 (high confidence). PPI network from STRING was visualized using the R package "networkD3" tool.

2. Second, protein degradation certainly leads to phenotypes in cells. It is recommended that the authors show the phenotypes of protein degradation with at least some of those degraders shown in this study.

Thank you for your valuable suggestion regarding the phenotypes resulting from protein degradation. To address this concern, we have conducted cell apoptosis, cell autophagy, and cell cycle assays to investigate the phenotypic effects of protein degradation using representative degraders.

In our previous manuscript, we have included the evaluation of apoptosis induced by DbTACs (Supplementary Fig. 13) and provided a detailed discussion of the findings in the Revised Manuscript Page 6 lines 359-363 with yellow background. The results demonstrate that DbTACs-26 Å induces significant cell apoptosis, particularly at the concentration of 200 nM. This characterization of apoptosis serves as an important phenotypic effect resulting from protein degradation induced by the degraders.

Additionally, we have performed a cell autophagy assay (Supplementary Fig. 15) to evaluate the phenotypic effects of protein degradation induced by the degraders. In this assay, MV4-11 cells were treated with various compounds, including the degraders (B11 and DbTACs), for a duration of 12 h. As a positive control, an autophagy inducer (Beyotime Biotechnology, Earle's Balanced Salt Solution) was applied for 4 h. The cells were then stained using the Autophagy Staining Assay Kit with MDC (Beyotime Biotechnology, C3018S) according to the provided instructions. The levels

of autophagy in MV4-11 cells were evaluated using a fluorescence microplate reader (SpectraMax® iD3), and the optical density (OD) values were measured at an excitation wavelength of 335 nm and an emission wavelength of 512 nm. Our results demonstrate that both B11 and all DbTACs groups induce autophagy, with the 200 nM concentration showing equivalent autophagy induction to the positive control. These measurements provide quantitative information on the extent of autophagy induction and allow us to assess the phenotypic effects of protein degradation. We have included the data and a detailed discussion of the observed phenotypic effects in the Revised Manuscript Page 6 lines 363-374 and in the Revised Supplementary information Page 10 lines 91-96 with yellow background. A detailed method of cell autophagy assay has been provided in the Revised Manuscript Page 13-14 lines 963-975 with yellow background.

Furthermore, in the cell cycle assay (Supplementary Fig. 20), MOLM13 cells were treated with bis-DbTACs at various concentrations (80, 200, 250, 500, and 1000 nM) or PBS (Control) for a duration of 12 h. After the treatment period, cells were harvested and fixed in ethanol overnight. The fixed cells were then stained using the Cell Cycle Detection Kit (Jiangsu KeyGEN Bio TECH Corp., Ltd, KGA, KGA512) following the manufacturer's protocol. Flow cytometry analysis was performed to determine the distribution of cells in different phases of the cell cycle, including G1, S, and G2 phases. The results indicate that the content of the S phase, which is responsible for DNA replication, decreases with increasing bis-DbTACs concentration, indicating the inhibition of cell proliferation. These findings further contribute to elucidating the phenotypic effects resulting from protein degradation induced by the degraders. We have included the discussion and results of the cell cycle assay in the Revised Manuscript Page 8 lines 462-470 and the Revised Supplementary information Page 12 lines 114-119 with yellow background. A detailed method of cell cycle assay has been provided in the Revised Manuscript Page 14 lines 976-985 with yellow background. Above supplementary data, discussion and method were attached as follows for your convenience.

By incorporating the phenotypic characterization of protein degradation induced by the degraders, we have enhanced the comprehensiveness and impact of our findings. We appreciate your valuable input and believe that these additions strengthen the overall significance and relevance of our research.

Supplementary Fig. 13. Cell apoptosis effect of DbTACs in MV4-11 cells. MV4-11 cells were treated with DbTACs-26 Å (80, 100, 150, 200, and 250 nM) for 6 h, then co-stained with annexin V/PI according to the instructions, and cell apoptosis was detected by flow cytometry. The annexin V/PI intensity dot plots showed the intensity of dot in the Q2 region increased significantly.

Supplementary Fig. 15. Activation of autophagy by DbTACs-26 Å treatment. Fluorescence microplate showing the levels of autophagy in MV4-11 cells following a 12-hour treatment with various compounds. An autophagy inducer (Earle's Balanced Salt Solution) was used as a positive control. OD values at excitation wavelength 335 nm and emission wavelength 512 nm were recorded. The error bars indicate the mean±SD values; n = 3.

Supplementary Fig. 20. Cell cycle analysis of bis-DbTACs. After a 12-hour treatment with PBS (Control) or the specified concentration of bis-DbTACs, MOLM13 cells were collected and fixed

overnight at 4°C and subjected to staining with PI/RNase A solution. Flow cytometric analysis was then performed to evaluate the cell cycle distribution. The reduction in the proportion of cells in the S phase indicated that bis-DbTACs led to impaired DNA synthesis. The FL2-A channel was used to measure PI fluorescence.

Please note that the changes have been marked in yellow in the revision:

After revision (lines 363-374): To evaluate the level of autophagy, we employed a fluorescence-based assay using an Autophagy Staining Assay Kit with MDC (Supplementary Fig. 15). After MV4-11 cells were treated with DbTACs-26 Å at different concentrations, the mean fluorescence intensity of the MDC probe was significantly increased compared with the control group, indicating the accumulation of acid compartments. In particular, the fluorescence intensity of the 200 nM DbTACs-26 Å was similar to that of the autophagy induced positive group, indicating that autophagosomes were formed during autophagy. Together, these findings demonstrate that DbTACs promote autophagy in our cellular model.

After revision (lines 963-975): Cell autophagy assay. MV4-11 cells were seeded into a 96-well blackboard and allowed to adhere overnight. Subsequently, the cells were treated with various compounds for a duration of 12 h. As a positive control, an autophagy inducer (Beyotime Biotechnology, Earle's Balanced Salt Solution) was applied for 4 h. After treatment, the cells were stained using the Autophagy Staining Assay Kit with MDC (Beyotime Biotechnology, C3018S) following the provided instructions. The fluorescence microplate (SpectraMax® iD3) was employed to evaluate the levels of autophagy in MV4-11 cells post-treatment. The optical density (OD) values were measured at an excitation wavelength of 335 nm and an emission wavelength of 512 nm.

After revision (lines 462-470): To assess the impact of bis-DbTACs on the cell cycle, we performed flow cytometry analysis employing propidium iodide (PI) staining (Supplementary Fig. 20). After treating MOLM13 cells with bis-DbTACs, a significant reduction in the proportion of cells in the S phase, which is critical for DNA replication, was observed when compared to the control group. This finding implied that bis-DbTACs possess the ability to impede DNA synthesis, thus modulating the cell cycle progression.

After revision (lines 976-985): Cell cycle assay. Briefly, MOLM13 cells were seeded in 6-well plates and treated with bis-DbTACs (final concentrations of 80, 200, 250, 500, and 1000 nM) or PBS (Control) for 12 h. After the treatment period, cells were harvested and fixed in ethanol. Fixed cells were then stained using the Cell Cycle Detection Kit (Jiangsu KeyGEN Bio TECH Corp., Ltd, KGA, KGA512) following the manufacturer's protocol. The flow cytometry data were processed using ImageJ software to determine the distribution of cells in different phases of the cell cycle, including G1, S, and G2 phases.

REVIEWERS' COMMENTS

Reviewer #1 (Remarks to the Author):

In this paper by Zhou et al., the authors describe a new technology using DNA origami framework as linkers for proteolysis targeting chimeras. In this re-submission, the authors answered all the question raised by the reviewers. The addition of a non-nuclear target as well as the added proteomic studies, really increase the quality of the paper. The authors now demonstrated clearly that their Db-TACs can enter cells and degrade a range of target proteins in different location. This paper should be of interest to the chemical biology community at large and especially to scientist working in the field of DNA origami / nucleic acids or targeted protein degradation. I fully support the publication of the manuscript in its current state.

Reviewer #2 (Remarks to the Author):

I have thoroughly reviewed the revised version, and the authors have made the requested revisions to the articles. I highly recommend the revised article for publication in Nature Communications.”

Reviewer #3 (Remarks to the Author):

In this study, the authors In this study, the authors have developed DNA framework-engineered chimeras platform to enable selectively targeted protein degradation, for which they named DbTACs. DNA tetrahedra were employed as templates for the synthesize PROTAC degraders by precisely locating ligands of POI and E3 ligase on templates, ligand spacings were controllably manipulated from 8 Å to 57 Å. A number of examples were provided to demonstrate the versatility of this approach. The authors showed that the ligands used for POIs can be small-molecule ligands, antibodies and DNAs. In the revised manuscript, the authors have done a great job in addressing all of the concerns raised in the original version. For example, the authors have done a global proteomics experiment to show that their designed BRD9 degrader indeed selectively down-regulates BRD9 protein with the highest significance, although a number of other proteins were also down- or up-regulated. They have also performed apoptosis and autophagy analyses with representative degraders to show that these degraders not only induce degradation of their POIs but also have expected phenotypes. The authors have also done an excellent job in addressing concerns raised by Reviewers #1 and #2. Overall, the revised manuscript has clearly demonstrated that the

DNA framework-engineered chimeras platform is very powerful in developing biological tools for target validation.

One important limitation with the current study is that the authors have not attempted to perform studies in animals. Therefore, it is unclear if DbTACs are capable of effectively inducing protein degradation in tissues (animals) and if DbTACs have any potential clinical applications. While these important limitations don't need to be addressed for the publication at this stage, they need to be clearly acknowledged in the final form of the manuscript.

Point-by-point Response to Reviewer Comments

We deeply appreciate the reviewer's comments on our manuscript (NCOMMS-23-02497B).

Reviewer 1

Comments:

In this paper by Zhou et al., the authors describe a new technology using DNA origami framework as linkers for proteolysis targeting chimeras. In this re-submission, the authors answered all the question raised by the reviewers. The addition of a non-nuclear target as well as the added proteomic studies, really increase the quality of the paper. The authors now demonstrated clearly that their Db-TACs can enter cells and degrade a range of target proteins in different location. This paper should be of interest to the chemical biology community at large and especially to scientist working in the field of DNA origami / nucleic acids or targeted protein degradation. I fully support the publication of the manuscript in its current state.

Response: Thank you for your positive feedback and comments on our manuscript. We express our gratitude for your support and recommendation for the publication of our manuscript in its current state.

Reviewer 2

Comments:

I have thoroughly reviewed the revised version, and the authors have made the requested revisions to the articles. I highly recommend the revised article for publication in Nature Communications.

Response: Thank you for your thorough review of the revised version of our manuscript. We greatly appreciate your high recommendation for the publication of our revised article in Nature Communications.

Reviewer: 3

Comments:

In this study, the authors have developed DNA framework-engineered chimeras platform to enable selectively targeted protein degradation, for which they named DbTACs. DNA tetrahedra were employed as templates for the synthesize PROTAC degraders by precisely locating ligands of POI and E3 ligase on templates, ligand spacings were controllably manipulated from 8 Å to 57 Å. A number of examples were provided to demonstrate the versatility of this approach. The authors showed that the ligands used for POIs can be small-molecule ligands, antibodies and DNAs. In the revised manuscript, the authors have done a great job in addressing all of the concerns raised in the original version. For example, the authors have done a global proteomics experiment to show that their designed BRD9 degrader indeed selectively down-regulates BRD9 protein with the highest significance, although a number of other proteins were also down- or up-regulated. They have also performed apoptosis and autophagy analyses with representative degraders to show that these degraders not only induce degradation of their POIs but also have expected phenotypes. The authors have also done an excellent job in addressing concerns raised by Reviewers #1 and #2. Overall, the revised manuscript has clearly demonstrated that the DNA framework-engineered chimeras

platform is very powerful in developing biological tools for target validation.

One important limitation with the current study is that the authors have not attempted to perform studies in animals. Therefore, it is unclear if DbTACs are capable of effectively inducing protein degradation in tissues (animals) and if DbTACs have any potential clinical applications. While these important limitations don't need to be addressed for the publication at this stage, they need to be clearly acknowledged in the final form of the manuscript.

Response: Thank you for your detailed evaluation of the revised version of our manuscript. We appreciate your positive comments and acknowledgment of the improvements we have made in response to the reviewers' comments.

We agree with your comment regarding the limitation of our study, specifically the absence of animal studies. To fully understand the potential, versatility, and generalizability of DbTACs technology, a more diverse range of target proteins and animal studies should be investigated in the future. Therefore, we have added a corresponding discussion in the Revised Manuscript Page 11 lines 649-653 with yellow background.

We are grateful for your thoughtful review and valuable feedback. Your comments have significantly contributed to the strengthening our manuscript.